# AMOR: A Recipe for Building Adaptable Modular Knowledge Agents Through Process Feedback

**Jian Guan**[1,2]**, Wei Wu**[2*]**, Zujie Wen**[2]**, Peng Xu**[2]**, Hongning Wang**[1]**, Minlie Huang**[1*]
[1]The CoAI group, DCST, Institute for Artificial Intelligence,
[1]State Key Lab of Intelligent Technology and Systems,
[1]Beijing National Research Center for Information Science and Technology,
[1]Tsinghua University, Beijing 100084, China. [2]Ant Group.
`{jianguanthu, wuwei19850318,wang.hongn}@gmain.com,`
`{zujie.wzj,peng.x}@antgroup.com, aihuang@tsinghua.edu.cn.`

## Abstract

The notable success of large language models (LLMs) has sparked an upsurge in building language agents to complete various complex tasks. We present AMOR, an agent framework based on open-source LLMs, which reasons with external knowledge bases and adapts to specific domains through human supervision to the reasoning process. AMOR builds reasoning logic over a finite state machine (FSM) that solves problems through autonomous executions and transitions over disentangled modules. This allows humans to provide direct feedback to the individual modules, and thus naturally forms process supervision. Based on this reasoning and feedback framework, we develop AMOR through two-stage fine-tuning: warm-up and adaptation. The former fine-tunes the LLM with examples automatically constructed from various public datasets, enabling AMOR to generalize across different knowledge environments, while the latter tailors AMOR to specific domains using process feedback. Extensive experiments across multiple domains demonstrate the advantage of AMOR to strong baselines, thanks to its FSM-based reasoning and process feedback mechanism. The code and data are publicly available at `https://github.com/JianGuanTHU/AMOR`.

## 1 Introduction

LLMs, with astounding performance over general natural language processing (NLP) problems [42, 1, 36], have spurred great interest in building LLM-based agents to solve complex tasks by interacting with external resources such as web knowledge [27], specialized tools [31], etc.

We focus on developing agents for knowledge-intensive tasks, where the agent completes users' information-seeking requests by interacting with specific knowledge bases [22]. To address the complexity of such tasks, we posit the desiderata for a qualifying agent as follows: Firstly, the agent should possess a robust *reasoning logic* about the task to solve individual problems with precise pathways. Secondly, the agent should maintain an *adaptive mechanism* to adjust to specific environments, rather than staying static. Thirdly, the reasoning process should be amenable to human interventions, enabling humans to steer the agent's behavior through direct *feedback* to the process rather than only to the outcome. This ability can significantly facilitate alignment between agent behavior and human intent [39].

Although extensive studies have been conducted on building language agents, few, if any, can fulfill all the required criteria due to their uncontrollable reasoning logic, static model capability, or

---

*Corresponding authors.

38th Conference on Neural Information Processing Systems (NeurIPS 2024).

Table 1: Comparison between AMOR and representative methods for building agents. Appendix A.1 provides a more comprehensive discussion in detail.

| Method | Reasoning Logic | | Adaptive Mechanism | Feedback |
|---|---|---|---|---|
| | Step | Inter-Step Dependency | | |
| **WebGPT** [27] | Tool Invoking | *Undefined* | Imitation Learning from Humans | Outcome |
| **CoT** [43] | Reasoning | *Undefined* | Prompting | *Undefined* |
| **ToT** [50] | Reasoning | *Undefined* | Prompting | Process |
| **ReAct** [51] | Reasoning&Tool Invoking | *Undefined* | Prompting | *Undefined* |
| **Reflexion** [35] | Reasoning&Tool Invoking | *Undefined* | Prompting | Process |
| **AgentLM** [53] | Reasoning&Tool Invoking | *Undefined* | Imitation Learning from LLMs | Outcome |
| **MetaGPT** [14] | Specialized Module | Sequential Pipeline | Prompting | Process |
| **LUMOS** [52] | Specialized Module | Sequential Pipeline | Imitation Learning from Humans | *Undefined* |
| **AMOR** | Specialized Module | Finite State Machine | Exploration&Exploitation | Process |

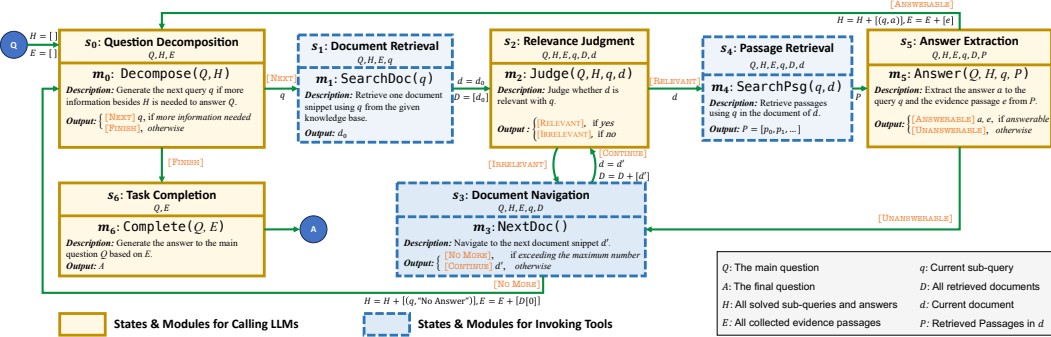

Figure 1: AMOR's state transition diagram. Each box represents a state and the corresponding module that is executed when entering the state. There may be multiple categories of execution results distinguished by special branch tokens such as "[NEXT]." Then AMOR determines the next state based on the branch tokens.

sparse/missing feedback signals, as detailed in Table 1. Consequently, it is still challenging for users to critique, and thus guide existing agents to follow targeted manners, especially when the agents are built upon less powerful LLMs [25].

We introduce **AMOR**, an **A**daptable **MO**dula**R** knowledge agent that can reason and adapt, with the reasoning process amenable to human supervision, based on open-source LLMs. AMOR's reasoning logic is formalized as a finite state machine (FSM) [7, 21] that solves problems via a series of executions and transitions over a set of modules (Figure 1). This naturally enables the desired process-based supervision mechanism, allowing users to give feedback to each LLM-controlled module. AMOR supports flexible forms of feedback, either binary judgments regarding the correctness or refinement of the outputs. The reasoning logic and process feedback mechanism together frame how AMOR thinks, acts, and interacts with users and task environments.

We build AMOR upon an LLM equipped with distinct parameters for different modules to efficiently handle multiple tasks. The training in AMOR happens in two stages: **(1) Warm-up:** the modular design enables us to construct training data separately for each disentangled module without requiring complete trajectories for specific tasks. As a result, we create a large dataset of 50k examples covering multiple distinct tasks, simply using public datasets. We fine-tune AMOR on this data for generalization over various knowledge-seeking scenarios. **(2) Adaptation:** when deployed, we tailor AMOR to the target domain by letting it autonomously address user tasks (i.e., exploration), collecting process feedback for each LLM output, and evolving through further fine-tuning on the exploration trajectories with feedback (i.e., exploitation). Our contributions are summarized as follows:

**I.** We propose a general framework for building knowledge agents, featuring FSM-based reasoning logic and a process feedback mechanism. We focus on text corpora as knowledge bases, but the approach can be flexibly extended to other knowledge types and user tasks by customizing the modules and dependencies within the FSM framework.

**II.** Experiments across multiple domains show the strong advantage of the FSM-based reasoning logic with 30%-40% improvements over baselines when based on off-the-shelf LLMs (e.g., GPT-4[2]). Switching to fine-tuned LLMs, the warm-up stage empowers AMOR to generalize to multiple domains and surpass strong baselines. After we adapt AMOR to specific domains, subsequent domain-specific adaptations reveal that process feedback is significantly more effective in improving the reasoning process than outcome feedback.

## 2  Related work

**Language agents.**   Interest is surging in building agents for tasks necessitating multi-step reasoning. Existing work falls into two groups. The first group focuses on designing agent architectures, such as CoT's step-by-step reasoning [44], ReAct's integration of reasoning, action, and observation to allow tool use [51], and CODEPLAN's two-stage reasoning framework that first generates a code-form plan and then realizes low-level reasoning steps [45]. Nevertheless, such free-form reasoning constraints human intervention. In contrast, modular agents follow a pipeline to execute specialized modules [19, 14, 11, 3, 52], improving the ease of intervention. The second group aims to design adaptive mechanisms for adapting agents to specific scenarios. ToT [50] and Reflexion [35] use environment feedback for multi-path pruning and iterative single-path refinement, respectively, but suffer from poor inference efficiency and need for real-time feedback. As a fine-tuning approach, recent work equipped open-source LLMs with specific agent abilities by learning from examples synthesized based on human priors [5], or expert trajectories from humans [27] or GPT-4 [53, 4] with correctness validation through outcome feedback. In contrast, our modular agent AMOR employs FSM-based reasoning with a stronger capacity for handling complex tasks than simple pipelines and adapts effectively to specific environments via process feedback.

**Retrieval-augmented generation (RAG).**   The RAG paradigm augments the inputs of LLMs with retrieved passages to enhance factuality [12, 22, 10]. Recent studies have developed interleaved reasoning-retrieval for better information recall than one-step retrieval [38, 16, 28]. However, retrieval may introduce noise that leads to low-quality answers [34]. To tackle this, Self-RAG [2] trained LLMs to selectively perform retrieval and utilize retrieved passages. Unlike RAG approaches, AMOR emphasizes an explainable reasoning process for proactively decomposing questions and seeking evidence for grounded generation, and allows for process feedback from humans. Nevertheless, RAG mainly focuses on integrating parametric factual knowledge in LLMs and retrieved non-parametric knowledge, which is less explainable and intervenable.

## 3  AMOR agent

AMOR relies on three key techniques: FSM-based reasoning logic, a process feedback mechanism, and a two-stage fine-tuning strategy. We detail the definition of the reasoning logic and its specification assuming the knowledge base is a text corpus in §3.1, the method for fine-tuning open-source LLMs as a warm-up stage in §3.2, and the adaptation stage driven by process feedback in §3.3.

---

**Algorithm 1** FSM-based Reasoning Logic

**Input:** Agent at the state $s = s_0$; $Q$: Question.
**Output:** $A$: Final Answer; $R$: Reasoning Process.
1  $R = [\,]$
2  **while** $s \neq s_{N-1}$ **do**
3      $y = m(s)$  // Obtain the output $y$ given $s$ from the corresponding module $m$.
4      $R$.append($\{$"state": $s$, "output": $y\}$)
5  $A = y$
6  **return** $A, R$

---

### 3.1  Reasoning logic

Algorithm 1 outlines how to deduce the answer $A$ for an input question $Q$ with a reasoning process $R$ using FSM-based reasoning logic, which can be defined by a quadruple: $\{\mathcal{S}, \mathcal{M}, \mathcal{E}, \mu\}$, where

- $\mathcal{S} = \{s_0, \ldots, s_{N-1}\}$ is a set of states with $s_0$ as the initial state and $s_{N-1}$ as the final state. Each state holds variables to track context information.
- $\mathcal{M} = \{m_0, \ldots, m_{N-1}\}$ is a set of modules with $m_k$ triggered when the reasoning flow reaches state $s_k$. The modules are categorized into two types: (a) Tool modules ($\mathcal{M}_{\text{TOOL}}$) for invoking tools, and (b) LLM modules ($\mathcal{M}_{\text{LLM}}$) for calling LLMs.

---

[2] In this work, GPT-3.5/4 refers to the OpenAI's API "gpt-3.5-turbo" / "gpt-4-1106-preview," respectively.

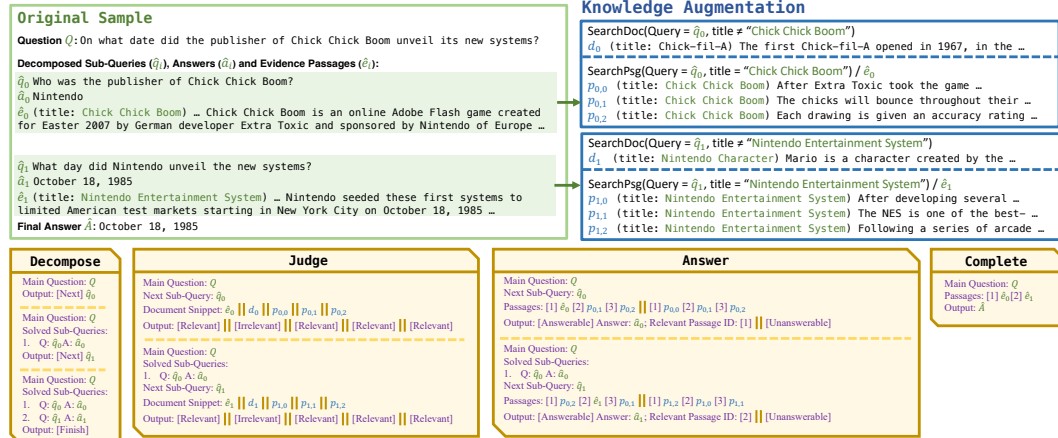

Figure 2: On the top left is a sample question from Musique [37], providing ample information (in **green**) for constructing training examples for four LLM modules of AMOR (bottom). We augment extra knowledge (in **blue**) for the Judge and Answer module by invoking the SearchDoc and SearchPsg tools (top right). In each example, we highlight the prompt in **purple** to format the current state (before "Output:") and output (after "Output:"), and use "||" to separate different examples for training.

- $\mathcal{E}$ is the set of all possible outputs of $\mathcal{M}$.
- $\mu : \mathcal{S} \times \mathcal{E} \to \mathcal{S}$ is the transition function that determines the next state of the reasoning flow given the current state and the execution result of the corresponding module.

When the external knowledge base is a text corpus, an instantiation of the reasoning logic can be represented by the state transition diagram in Figure 1. In this case, $\mathcal{M}_{\text{TOOL}}$ perform document and passage retrieval using external retrievers; while $\mathcal{M}_{\text{LLM}}$ leverage the LLM to analyze and digest the question, documents, and passages to deduce the final answer. To distinguish different types of outputs from a module that requires different subsequent modules, we employ a set of special branch tokens such as "[NEXT]" to guide $\mu$ to determine the next state. In summary, AMOR answers question $Q$ by **(1)** iteratively decomposing $Q$ to a sub-query $q$ at state $s_0$, and finding the answer $a$ to $q$ and the evidence passage $e$ through iterative knowledge retrieval, relevance evaluation, retrieval refinement (i.e., "Passage Retrieval"), and answer extraction, until no more knowledge is needed; and **(2)** deducing the final answer $A$ based on the collected evidence passages at the final state.

Defining reasoning logic as an FSM offers three advantages: **(1) Structured Thinking.** FSM makes specifications of inter-step dependencies (e.g., prioritization, branch selection) easy, and thus enables narrowing down the exploration space. **(2) Skill Disentanglement.** By decomposing complex tasks into modular steps, one can independently construct training data for each module, which significantly reduces the difficulty of implementing AMOR with open-source LLMs (cf., §3.2). This feature also allows AMOR to focus on single steps, thereby mitigating the weakness of LLMs in reasoning over long context formed by task-solving trajectories [24]. **(3) Intervenable Workflow.** The structured reasoning process enables users to easily diagnose the agent's mistakes and provide process feedback for improving the reasoning capability of the agent (§3.3).

## 3.2 Warming-up open-source LLMs

Open-source LLMs are observed to fall short in complex agent tasks [46, 25]. Recent studies have improved their reasoning abilities through imitation learning using trajectories from advanced LLMs such as GPT-4 [53, 4]. However, even GPT-4 can struggle with producing high-quality reasoning trajectories [29].

AMOR's modular design enables us to construct training data for each module separately from existing datasets without simulating the whole trajectories, thus greatly alleviating the above issue. Formally, given a sample question $Q$ with annotations of the final answer $\hat{A}$, all sub-queries and answers $\hat{H} = [(\hat{q}_0, \hat{a}_0), (\hat{q}_1, \hat{a}_1), \cdots]$, and all evidence passages $\hat{E} = [\hat{e}_0, \hat{e}_1, \cdots]$, we can directly

transform these annotations into a suitable format to serve as training data for Decompose and Complete in Figure 1. Since Judge and Answer require multiple types of retrieved knowledge (e.g., *relevant* or not), we employ retrieval tools to augment the input. Figure 2 exemplifies the construction pipeline, which can be easily extended to other knowledge-intensive datasets and specific domains. Appendix A.4 shows more details.

When fine-tuning open-source LLMs to handle multiple tasks defined by different modules, we are inspired by the Mixture-of-Experts approach [33] to learn distinct Feed-Forward Network (FFN) parameters in the final quarter of the Transformer blocks to balance the trade-off between performance and inference efficiency. These module-specific parameters are initialized using the original model's FFN layers. We call the proposed architecture **M**odule-**A**ware **M**ixture-**o**f-**E**xperts (MA-MoE)[3]. Then, we fine-tune the MA-MoE model with the standard language modeling loss:

$$\mathcal{L}_1 = -\mathbb{E}_{m \in \mathcal{M}_{\text{LLM}}, (\hat{s}, \hat{y}) \in \mathcal{D}_m} \lambda_m \log \pi_{\theta_m}(\hat{y}|\hat{s}), \tag{1}$$

where $\pi$ refers to the policy model MA-MoE that maps the state $\hat{s}$ to an action $\hat{y}$, $\theta_m$ denotes the parameter for the module $m \in \mathcal{M}_{\text{LLM}}$, $\mathcal{D}_m$ is the corresponding collection of training examples, $(\hat{s}, \hat{y})$ is a state-output pair from $\mathcal{D}_m$, and $\{\lambda_m\}$ are tunable hyper-parameters.

### 3.3 Adaptation through process feedback

Feedback is crucial for adapting language agents to specific environments [40], especially when dealing with unseen, long-tail, or ever-changing domain knowledge. Prior agents commonly used outcome feedback for adaptation which assesses the correctness of intermediate steps based on the success or failure of the outcome [53, 4]. However, outcome feedback is too sparse to improve intermediate reasoning [23]. Recent studies also highlighted that LLMs' reasoning steps are likely to contradict the outcome [26], which means that outcome feedback may inevitably introduce noise during training (see examples in Appendix B.8). In contrast, AMOR's process feedback mechanism can effectively alleviate these issues.

---

**Algorithm 2** Adaptation through Process Feedback

**Input:** $\{\pi_{\theta_m}^{\text{WFT}}\}$: Initial Policy; $T$: Exploration Steps between Exploitation; $I$: Number of Iterations.
**Output:** $\{\pi_{\theta_m}\}$: Adapted Policy.

1 **while** $i \leftarrow 1$ **to** $I$ **do**
2      $\mathcal{R} = [\,]$ // Feedback-Refined Reasoning Processes
3      **while** $t \leftarrow 1$ **to** $T$ **do**
         // Exploration
4          Receive an input question $Q$.
5          Collect AMOR$_\theta$'s reasoning process $R$. // Algorithm 1
         // Feedback Collection for Each LLM Module
6          **foreach** Step $r_k \in R$ $(k = 0, 1, 2, \cdots)$ **do**
7              Extract the state $s_k$ and output $y_k$ from $r_k$.
8              **if** The corresponding module $m_k \in \mathcal{M}_{\text{LLM}}$ **then**
9                  Collect feedback $f_k$ for $s_k$ and $y_k$.
10                  Determine $\tilde{y}_k$ and $o_k$ based on $f_k$. // Eq. 2
11                  $\mathcal{R}$.append($[s_k, \tilde{y}_k, o_k]$)
     // Exploitation
12      Optimize $\{\theta_m\}$ to minimize $\mathcal{L}_2$ on $\mathcal{R}$. // Eq. 3
13 **return** $\{\pi_{\theta_m}\}$

---

Algorithm 2 describes the adaptation mechanism of AMOR parameterized by $\theta$, specifically as three steps: **(1) Exploration.** AMOR answers the input question $Q$ by interacting with a knowledge base. **(2) Feedback Collection.** AMOR's reasoning process for $Q$ is evaluated with feedback $f_k$ for the output $y_k$ of the LLM at each step during reasoning, which is either "right/wrong" or a refined version of $y$. We convert $y$ into a feedback-refined target output $\tilde{y}$ based on the feedback $f_k$ and determine the immediate reward $o_k$ as follows:

$$\tilde{y}_k, o_k = \begin{cases} y_k, 1 & \text{if } f_k = \text{``right''}, \\ y_k, 0 & \text{if } f_k = \text{``wrong''}, \\ f_k, 1 & \text{if } f_k \text{ is refinement.} \end{cases} \tag{2}$$

**(3) Exploitation.** Every $T$ steps of the former exploration and feedback collection, we optimize the initial policy based on the resulting trajectories and corresponding feedback [32]:

$$\mathcal{L}_2 = -\mathbb{E}_{m \in \mathcal{M}_{\text{LLM}}, (s_k, \tilde{y}_k, o_k) \in \mathcal{R}_m} \lambda_m [o_k - \beta \log(\pi_{\theta_m}(\tilde{y}_k|s_k) / \pi_{\theta_m}^{\text{WFT}}(\tilde{y}_k|s_k))], \tag{3}$$

---

[3] "Module-Aware" means when AMOR executes a certain module, its module index will be provided to the routers of the model to indicate which expert should be activated.

Table 2: Automatic annotation strategy for silver process feedback for different LLM modules.

| Module $m$ | Output $y$ | Silver Process Feedback $f$ |
|---|---|---|
| **Decompose**$(Q, H)$ | [NEXT] $q$ | "right", if the retrieved documents using $q$ overlap the documents corresponding to $\hat{E}$; "wrong", otherwise. |
| | [FINISH] | "right", if $\hat{E} \subseteq E$ (i.e, evidence passages collected by AMOR); "wrong", otherwise. |
| **Judge**$(Q, H, q, d)$ | [RELEVANT] [IRRELEVANT] | "[RELEVANT]", if one of passages in $\hat{E}$ comes from the same document as $d$; "[IRRELEVANT]", otherwise. |
| **Answer**$(Q, H, q, P)$ | [ANSWERABLE] $a$ $e$ | "right", if $e \in \hat{E}$; "wrong", otherwise |
| | [UNANSWERABLE] | "right", if $P \cap \hat{E} = \emptyset$; "wrong", otherwise |
| **Complete**$(Q, E)$ | $A$ | $\hat{A}$, if $\hat{E} \subseteq E$; "wrong", otherwise. |

where $\mathcal{R}_m \subseteq \mathcal{R}$ denotes the training examples for module $m$, $\pi_\theta^{\text{WFT}}$ refers to the initial warm-up policy. Notably, this loss function is non-differentiable, necessitating the use of a specialized optimization technique. We use a recently proposed alignment algorithm KTO [9] with an MLE regularization [47] for optimization, which optimizes the policy without requiring paired human preferences. Crucially, when optimizing a particular module $m$, the gradient induced by the feedback signal propagates through the entire MA-MoE model, except for the FFN layers corresponding to other modules. This targeted optimization approach enables AMOR to effectively align its outputs with the desired intermediate results and final answers, leveraging the fine-grained process feedback provided by human supervisors.

## 4 Experiments

### 4.1 Experimental setup

**Tools modules.** We construct retrievers for both SearchDoc and SearchPsg using Contriever-MS MARCO [15]. SearchDoc retrieves a single document snippet per query, while SearchPsg fetches the top three relevant passages from a given document. By invoking NextDoc, at most nine more document snippets are returned. Appendix B.1 presents more details.

**Warm-up datasets.** We employ four question-answering (QA) datasets to warm up open-source LLMs, including 2WikiMultiHopQA [13], Musique [37], NaturalQuestions [20] and BoolQ [6]. They vary in levels of question complexity (single- or multi-hop), answer types (phrase spans or yes/no), and types of dependency structures between sub-queries (e.g., serial or parallel), etc. Appendix A.4 shows the statistics in detail.

**Adaptation & evaluation datasets.** We consider three benchmarks, by which we simulate different deployment scenarios: **(1) HotpotQA** [49]: a challenging multi-hop QA dataset built on Wikipedia articles. We use the Wikipedia dump provided in [15] as the knowledge base. **(2) PubMedQA** [17]: a biomedical QA dataset that requires answering a question by "yes/no" given a PubMed abstract. We adapt the data to retrieval-based QA by piling all 274k abstracts provided in the paper as a knowledge base, where each document comprises one abstract passage. **(3) QASPER** [8]: answering questions in free form based on a long NLP paper. For each question, we regard the corresponding paper as a knowledge base and each section of the paper as a document with several passages. We use the training and validation sets for adaptation fine-tuning and the test sets for evaluation. For evaluation metrics, we use exact match (EM) and F1 scores for HotpotQA and QASPER; and the accuracy (ACC) of "yes/no" for PubMedQA. More details are in Appendix B.2.

**Feedback annotation.** Considering limited resources, we simulate human behavior and provide silver feedback to AMOR's reasoning processes based on the gold answer $\hat{A}$ and gold evidence passages $\hat{E} = [\hat{e}_0, \hat{e}_1, \cdots]$ for each target question $Q$, which are already included in the training and validation data of the three benchmarks. Table 2 shows how we annotate the feedback for each LLM output $y$. Note that AMOR is applicable for gold feedback from humans in realistic applications. Appendix B.3 discusses the accuracy of the silver feedback through human evaluation.

**Implementation details.** We set $\lambda_m$ in Eq. 1 and Eq. 3 to 1 for all modules, $I = 1$ in Algorithm 2, and $T$ to the size of the training set for each dataset, and fine-tune LLAMA-2-7B/13B-Chat for two epochs with a learning rate of $2\mathrm{e}^{-5}$ using 8 NVIDIA 80GB A100 GPUs. While applying

Table 3: Refining each module output $y$ to $\tilde{y}$ based on the outcome feedback $f_o$ to adapt AMOR, where $\neg y$ denotes converting the binary output $y$ to its opposite label.

| Module $m$ | Target Output $\tilde{y}_k$ and Immediate Reward $o_k$ |
|---|---|
| **Decompose**$(Q, H)$ | $\tilde{y}_k = y$ and $o_k = 1$ if $f_o = \hat{A}$; $\tilde{y}_k = y$ and $o_k = 0$, otherwise. |
| **Judge**$(Q, H, q, d)$ | $\tilde{y}_k = y$ and $o_k = 1$, if $f_o = \hat{A}$; $\tilde{y}_k = \neg y$ and $o_k = 1$, otherwise. |
| **Answer**$(Q, H, q, P)$ | $\tilde{y}_k = y$ and $o_k = 1$ if $f_o = \hat{A}$; $\tilde{y}_k = y$ and $o_k = 0$, otherwise. |
| **Complete**$(Q, E)$ | $\tilde{y}_k = \hat{A}$ and $o_k = 1$ if $\hat{E} \subseteq E$; $\tilde{y}_k = y$ and $o_k = 0$, otherwise. |

AMOR for inference, we use greedy decoding for all generations. Besides, we set the maximum number of decomposed sub-queries to the maximum count of gold evidence passages, i.e., $2/1/1$ for HopotQA/PubMedQA/QASPER, respectively. Once the maximum number is reached, AMOR is transited to state $s_6$ ("Task Completion") to finalize the answer.

**Baselines.** We compare AMOR to various baselines with or without fine-tuning: **(1) CoT** [44]: it prompts an off-the-shelf LLM to generate the answer through step-by-step reasoning. **(2) RAG**: One-Step Retrieval (OneR) uses the question as a query to retrieve top-$K$ document snippets with the SearchDoc module to augment the input. We set $K$ as the maximum number of gold evidence passages in each dataset. Under the fine-tuning setting, we use the gold evidence passages for training. Self-RAG [2] selectively performs retrieval and utilizes retrieved passages while does not explicitly introduce question decomposition. They can be viewed as simplifications of AMOR. **(3) ReAct** [51]: it interleaves thought, action, and observation steps. An action can be either invoking the retrieval tools or finalizing an answer. We also compare AMOR with fine-tuned ReAct-style agents including AgentLM [53] and FIREACT [4]. We set the maximum number of action steps to 20. **(4) Modular Agents:** ReWoo [46] follows a pipeline that plans all sub-goals, generates actions, and then executes, while LUMOS [52] applies this pipeline iteratively, tackling one sub-goal at a time with each interaction. Both agents utilize GPT-3.5 as a supplementary QA tool during action generation. Similar to AMOR, they modularize language agents; however, they lack explicit mechanisms for assessing the relevance of retrieved information. Under the setting without fine-tuning, we provide in-context examples for the baselines following their official implementations.

Furthermore, we also conduct ablation studies to investigate the influence of different components, resulting in two more baselines: **(1)** $\text{AMOR}_{\text{WFT}}$: AMOR with only warm-up fine-tuning, without further adaptation; and **(2)** $\text{AMOR}_{\text{Outcome}}$: outcome feedback instead of process feedback is utilized in adaptation after AMOR is warmed-up. Specifically, we determine the target output and corresponding immediate reward for an LLM module as detailed in Table 3, and then adapt AMOR using Eq. 3. For clarity, we denote our final method as $\text{AMOR}_{\text{Process}}$.

## 4.2 Main results

Table 4 reports the evaluation results of AMOR and baselines on three datasets, revealing three key findings: **(1) The FSM paradigm is clearly advantageous to prior agent frameworks.** $\text{AMOR}_{\text{w/o FT}}$ delivers strong performance by improving 41.9%, 32.1%, and 41.2% over ReAct on average when built on top of off-the-shelf LLMs, including L-7B, GPT-3.5, and GPT-4, respectively. This indicates that our proposed FSM paradigm is more effective in leveraging LLMs for complex reasoning. **(2) Warm-up fine-tuning generally enhances AMOR in downstream tasks.** When based on L-7B, $\text{AMOR}_{\text{WFT}}$ outperforms $\text{AMOR}_{\text{w/o FT}}$ across all datasets. Furthermore, $\text{AMOR}_{\text{WFT}}$ also surpasses other fine-tuned ReAct-style and modular agents, even including FIREACT that is fine-tuned with in-domain HotpotQA trajectories from GPT-4. This suggests the potential of utilizing existing datasets for developing powerful agents with well-defined reasoning logic. **(3) Process feedback is more effective than outcome feedback in facilitating the adaptation of agents.** The order that $\text{AMOR}_{\text{Process}} > \text{AMOR}_{\text{Outcome}} > \text{AMOR}_{\text{WFT}}$ indicates the impact of feedback in terms of tailoring agent behavior to specific domains, and process feedback is more helpful than outcome feedback for leading to the correct final answers.

Additionally, Table 5 presents the results from employing various model architectures (including our proposed MA-MoE model, the standard MoE model, and the Transformer model) as well as optimization algorithms (namely KTO and Supervised Fine-Tuning, i.e., SFT). The MoE model is identical to

Table 4: Results of AMOR and baselines. "L-7/13B" is short for "LLAMA-2-7/13B-Chat." We highlight the best results in **bold** and underline the second best. Models marked with $*$ are fine-tuned on the target datasets. Results marked with $\dagger$ are reported in the original paper and those marked with $\ddagger$ are reported in [4]. *N/A* means the method does not apply to the datasets. AMOR$_{\text{Process}}$ outperforms baselines under the same setting significantly ($p < 0.01$, sign test).

| Method | Base LLM | HotpotQA | | PubMedQA | QASPER | |
| --- | --- | --- | --- | --- | --- | --- |
| | | EM | F1 | ACC | EM | F1 |
| **Without Fine-Tuning** | | | | | | |
| ReAct | **L-7B** | 12.2 | 16.6 | 61.8 | 6.0 | 19.2 |
| AMOR$_{\text{w/o FT}}$ | **L-7B** | 26.0 | 34.6 | 62.9 | 4.5 | 21.3 |
| CoT | **GPT-3.5** | 28.0$^\ddagger$ | - | *N/A* | *N/A* | *N/A* |
| OneR | **GPT-3.5** | 33.4 | 42.1 | 72.6 | 6.8 | 23.3 |
| ReAct | **GPT-3.5** | 30.8 | 38.8 | 58.2 | 5.8 | 27.0 |
| ReWoo | **GPT-3.5** | 30.4$^\dagger$ | 40.1$^\dagger$ | - | - | - |
| AMOR$_{\text{w/o FT}}$ | **GPT-3.5** | 39.6 | 49.3 | 68.8 | 10.0 | 30.8 |
| CoT | **GPT-4** | 45.0$^\ddagger$ | - | *N/A* | *N/A* | *N/A* |
| ReAct | **GPT-4** | 42.0$^\ddagger$ | - | 62.1 | 7.1 | 26.2 |
| AMOR$_{\text{w/o FT}}$ | **GPT-4** | **55.2** | **65.2** | **80.0** | **11.5** | **37.4** |
| **With Fine-Tuning** | | | | | | |
| OneR$^*$ | **L-7B** | 34.8 | 43.8 | 75.3 | 11.0 | 25.5 |
| Self-RAG | **L-7B** | 22.4 | 32.9 | 62.6 | 2.1 | 17.9 |
| AgentLM | **L-7B** | 22.0$^\dagger$ | - | 64.9 | 4.2 | 20.2 |
| FIREACT | **L-7B** | 26.2$^\dagger$ | - | 66.1 | 6.5 | 18.4 |
| LUMOS | **L-7B** | 29.4$^\dagger$ | - | 70.3 | 7.1 | 19.5 |
| AMOR$_{\text{Process}}$$^*$ | **L-7B** | 45.8 | 54.9 | 81.1 | **19.1** | 35.3 |
| AMOR$_{\text{WFT}}$ | **L-7B** | 33.6 | 41.9 | 73.4 | 11.1 | 23.6 |
| AMOR$_{\text{Outcome}}$$^*$ | **L-7B** | 40.8 | 49.3 | 77.5 | 9.4 | 25.4 |
| AgentLM | **L-13B** | 29.6$^\dagger$ | - | 67.9 | 7.1 | 24.4 |
| AMOR$_{\text{Process}}$$^*$ | **L-13B** | **48.6** | **55.3** | **82.2** | 18.1 | **38.0** |
| AMOR$_{\text{WFT}}$ | **L-13B** | 36.8 | 44.1 | 74.6 | 15.2 | 27.3 |
| AMOR$_{\text{Outcome}}$$^*$ | **L-13B** | 42.4 | 51.6 | 80.1 | 9.9 | 26.5 |

Table 5: Results of AMOR$_{\text{Process}}$ based on L-7B with different architectures and optimization algorithms. The architecture setting is also applied on the warm-up fine-tuning stage. $\dagger$ refers to our final method.

| Architecture | Optimization | HotpotQA | | PubMedQA | QASPER | |
| --- | --- | --- | --- | --- | --- | --- |
| | | EM | F1 | ACC | EM | F1 |
| **MA-MoE**$^\dagger$ | **KTO**$^\dagger$ | **45.8** | **54.9** | **81.1** | **19.1** | **35.3** |
| **MA-MoE** | **SFT** | 43.2 | 53.1 | 79.3 | 18.6 | 34.2 |
| **MoE** | **SFT** | 41.6 | 51.1 | 78.8 | 17.5 | 33.5 |
| **Transformer** | **SFT** | 41.4 | 50.9 | 78.2 | 17.8 | 33.2 |

the MA-MoE model except it lacks module-specific awareness. SFT refers to optimizing the model only for outputs $\tilde{y}_k$ that receive an immediate reward $o_k = 1$, using standard language modeling loss. The results show: (1) The standard MoE architecture struggles to effectively differentiate its experts for multitasking scenarios, leading to performance comparable to the Transformer model. In contrast, the MA-MoE's module-specific awareness enables it to handle diverse tasks within the agent more adeptly. (2) KTO outperforms SFT in aligning the agent's performance with external feedback, owing to its exploitation of negative samples.

## 4.3 Discussions

The main results have substantiated the benefits of different components of AMOR for successfully completing tasks. Nonetheless, we are still curious about four key research questions: **(1) RQ1:**

Table 6: Recall scores under different settings.

| Method | Base LLM | HotpotQA | PubMedQA | QASPER |
|---|---|---|---|---|
| **OneR** | **N/A** | 31.1 | 67.6 | 24.9 |
| $\textbf{AMOR}_{\text{w/o FT}}$ | **L-7B** | 24.1 | 54.2 | 24.3 |
| $\textbf{AMOR}_{\text{Process}}$ | **L-7B** | 53.5 | 79.8 | 41.9 |
| $\textbf{AMOR}_{\text{WFT}}$ | **L-7B** | $\overline{41.1}$ | $\overline{69.2}$ | $\overline{27.5}$ |
| $\textbf{AMOR}_{\text{Outcome}}$ | **L-7B** | 40.2 | 70.0 | 27.5 |
| $\textbf{AMOR}_{\text{Process}}$ | **L-13B** | **53.7** | **80.5** | **42.4** |
| $\textbf{AMOR}_{\text{WFT}}$ | **L-13B** | 43.0 | 69.4 | 27.0 |
| $\textbf{AMOR}_{\text{Outcome}}$ | **L-13B** | 41.5 | 68.1 | 27.7 |

How do the AMOR variants differ in the ability to collect evidence? **(2) RQ2:** Is process feedback more data-efficient than outcome feedback for adaptation? **(3) RQ3:** What if using human feedback for adaptation of AMOR? **(4) RQ4:** To what extent does feedback-driven adaptation enhance the AMOR's reasoning process? Besides, Appendix B.6 and B.7 further demonstrate the efficient token usage of AMOR and the flexibility of AMOR's reasoning framework, respectively.

**RQ1: Evidence collection comparison.** We use recall of gold evidence passages ($\hat{E}$) among those collected by AMOR ($E$) to assess AMOR's ability to collect evidence, formally as $\frac{\#\{\hat{E} \cap E\}}{\#\{\hat{E}\}}$.

As shown in Table 6, we observe: **(1)** Warm-up fine-tuning consistently enhances evidence collection, with $\text{AMOR}_{\text{WFT}}$ achieving higher recall than $\text{AMOR}_{\text{w/o FT}}$ across all datasets. **(2)** Adaptation through outcome feedback ($\text{AMOR}_{\text{Outcome}}$) exerts a negligible impact on the recall results compared with $\text{AMOR}_{\text{WFT}}$, suggesting the superiority of $\text{AMOR}_{\text{Outcome}}$ to $\text{AMOR}_{\text{WFT}}$ in final answers (see Table 4) may stem from the improvement of Complete. **(3)** Process feedback is crucial to improve the evidence collection ability, with $\text{AMOR}_{\text{Process}}$ substantially outperforming the other variants.

**RQ2: Data efficiency for adaptation.** We aim to compare the data efficiency of different feedback types for adaptation in terms of the number of exploratory instances required. To this end, we adjust the exploration steps $T$ in Algorithm 2, selecting values at intervals of 200, ranging up to 2,000 steps on HotpotQA. Appendix B.5 further discusses the cases with $I > 1$ in Algorithm 2 where AMOR is optimized over multiple rounds.

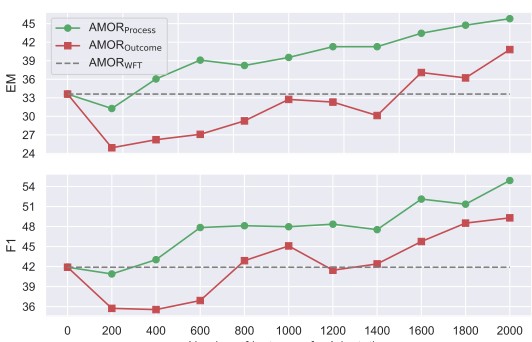

Figure 3: EM/F1 on HotpotQA varying with the number of exploratory instances for adaptation.

Figure 3 shows the post-adaptation performance of AMOR varying with the number of exploratory instances (i.e., $T$). Compared to $\text{AMOR}_{\text{Outcome}}$, $\text{AMOR}_{\text{Process}}$ requires significantly fewer exploration steps to achieve comparable performance. Notably, $\text{AMOR}_{\text{Outcome}}$ shows a marked decline in performance when exposed to a limited number of exploratory instances ($< 800$), suggesting a reduced adaptability in exploration-limited scenarios. Conversely, $\text{AMOR}_{\text{Process}}$'s robust performance under such constraints highlights its superior adaptability and efficiency with minimal interaction.

**RQ3: Adaptation through human feedback** Due to limited resources, we use automatically annotated silver feedback as a proxy for gold human feedback in our main experiments. We would like to emphasize that our experimental framework is inherently designed to seamlessly incorporate human feedback in place of its automated counterpart. To illustrate this, we carry out a human study to demonstrate how AMOR is adapted through human feedback on HotpotQA. For this study, we hire an NLP expert to provide human feedback for each module within $\text{AMOR}_{\text{WFT}}$ on 2,000 exploratory

Table 7: Adaptation results of AMOR through human feedback on HotpotQA based on L-7B.

| Agents | Feedback Type | EM | F1 |
|---|---|---|---|
| **AMOR**$_{\text{Process}}$ | Automatic Feedback | 45.8 | 54.9 |
| **AMOR**$_{\text{Process}}$ | Human Feedback | 50.8 | 59.2 |

Table 8: Accuracy of four LLM modules. All AMOR variants are based on L-7B.

| Method | Decompose | Judge | Answer | Complete |
|---|---|---|---|---|
| **AMOR**$_{\text{Process}}$ | **73.0** | **97.2** | **82.5** | **50.0** |
| **AMOR**$_{\text{WFT}}$ | 59.5 | 95.3 | 77.2 | 32.0 |
| **AMOR**$_{\text{Outcome}}$ | 61.2 | 96.0 | 75.1 | 44.0 |

instances following the annotation strategy in Appendix B.3. Table 7 shows the adaptation results using the collected human feedback.

The results distinctly suggest: human feedback more effectively adapts AMOR to specific knowledge environments than automatic feedback. Our study lays a robust groundwork for the practical deployment and real-world utilization of the AMOR framework.

**RQ4: Reasoning process assessment.** To measure the accuracy of AMOR's reasoning process, we performed a human study on HotpotQA, which involved: (1) selecting 50 random questions; (2) manually annotating the gold feedback $f_{\text{human}}$ for each LLM module output the following instructions using the same annotation protocol in Appendix B.3; and (3) calculating the accuracy of each LLM module output based on $f_{\text{human}}$ (1/0 indicating "right/wrong").

Table 8 presents the accuracy of AMOR variants, affirming RQ1's findings: process feedback significantly improves the reasoning process over AMOR$_{\text{WFT}}$ that lacks adaptation, while outcome feedback has a negligible effect. Moreover, AMOR$_{\text{Process}}$ relatively lags in the Decompose and Complete modules, hinting that future enhancements could focus on including more corresponding data during two fine-tuning stages.

### 4.4 Case study

Appendix B.8 presents several examples to further illustrate AMOR's strengths in reasoning logic and intervenability, as well as the limitations of relying on outcome feedback for adaptation, emphasizing the crucial role of process feedback.

## 5 Conclusion

In this work, we develop AMOR, an adaptable modular agent designed for knowledge-intensive tasks, featuring FSM-based reasoning logic and a process feedback mechanism. Based on open-source LLMs, AMOR undergoes a two-stage fine-tuning: initial warm-up to generalize across task environments and subsequent domain-specific adaptation through process feedback. Extensive experiments demonstrate AMOR's advantages over strong baselines across multiple domains. Further discussions highlight the effectiveness and efficiency of process feedback in adaptation. compared to previous agents. Future work will explore extending our paradigm to more knowledge types (e.g., structured knowledge bases) and broader agent tasks, ultimately empowering LLMs to autonomously design FSM-based reasoning logic on top of our paradigm.

## 6 Acknowledgements

We thank the anonymous reviewers and area chairs for their valuable feedback and insightful comments that helped improve this work. This work was supported by the NSFC projects(Key project with No. 61936010). This work was supported by the National Science Foundation for Distinguished Young Scholars (with No. 62125604).

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

# A Methodology

## A.1 Comparing AMOR with related works in detail

As illustrated in Figure 4, AMOR addresses the issues of prior reasoning methods in terms of three aspects:

- **AMOR is equipped with a controllable FSM-based reasoning logic with a stronger capacity for handling complex tasks than simple pipelines employed by Self-RAG, ReWoo, LUMOS.** For instance, if no relevant passages are retrieved from a document, AMOR can dynamically transit to the next document, while LUMOS would be constrained to generate answers based on the irrelevant passages, potentially leading to incorrect or low-quality outputs.
- **AMOR adapts to new environments through exploration and exploitation.** AMOR is designed with an adaptation fine-tuning stage, enabling it to adapt effectively to specific domains based on human feedback. This adaptive mechanism sets AMOR apart from prior modular agents that lack the ability to incorporate expert guidance.
- **AMOR enables humans to conveniently and effectively intervene and provide feedback at each reasoning step.** AMOR introduces a process feedback mechanism that enables humans to provide direct feedback on the individual modules within the FSM-based reasoning process. This approach facilitates a more natural and interpretable form of supervision, allowing for targeted improvements and fine-tuning of specific reasoning components.

In summary, AMOR achieves more controllable, adaptable, and human-guided reasoning capabilities compared to existing methods.

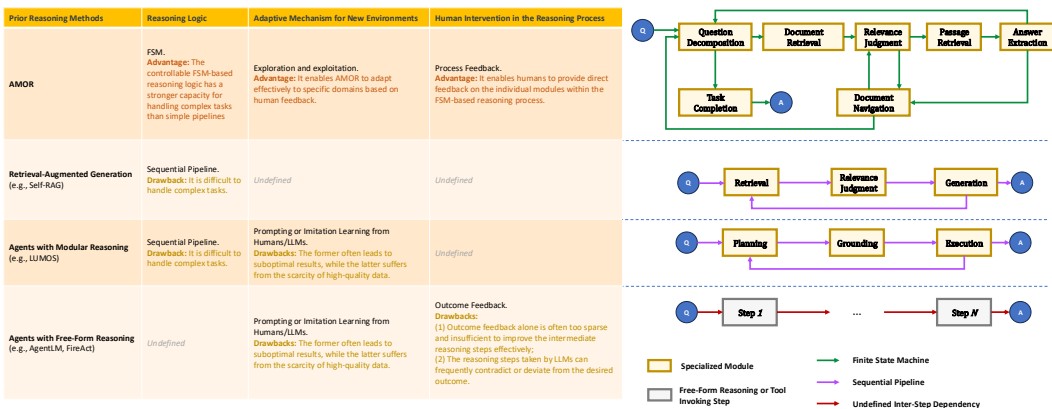

Figure 4: **Left:** Elaboration regarding the advantages and drawbacks when comparing AMOR with prior agents in terms of three aspects in Table 1. **Right:** The reasoning processes of AMOR and related works.

## A.2 Full algorithm of AMOR

Algorithm 2 in the main paper illustrates a general FSM-based reasoning logic, which can be adapted to various agent environments by defining the FSM including the states, modules, etc.

As shown in Algorithm 3, AMOR provides an instantiation of the FSM-based reasoning logic for the knowledge-seeking scenarios following the state transition diagram in Figure 1 in the main paper. We expect to extend this work to more environments in the future.

## A.3 Prompts for LLM modules

Table 10, 11, 12 and 13 show the prompts for four LLM modules in AMOR under the "Without Fine-tuning" setting on HotpotQA. They can be converted to the "With Fine-tuning" setting by removing the in-context examples. The prompts for PubMedQA and QASPER are similar.

**Algorithm 3** Answering Question $Q$ Using AMOR

**Data:** AMOR at the initial state $s = s_0 (Q, H, E)$; $Q$: Question; $H = [\,]$: All solved sub-queries and answers; $E = [\,]$: All evidence passages collected by AMOR.
**Result:** $A$: Final Answer; $R$: Reasoning Process.

```
1  while True do
2      if s = s₀ then
3          y = Decompose(s.Q, s.H)
4          R.append({"state": s, "output": y})
           //  Transit to the next state.
5          if y starts with "[NEXT]" then
6              Extract the next sub-query q from y
7              s = s₁(s.Q, s.H, s.E, q)
8          else if y starts with "[FINISH]" then
9              s = s₆(s.Q, s.E)
10     else if s = s₁ then
11         y = SearchDoc(s.q)
           //  Transit to the next state.
12         D, d = [y], y
13         s = s₂(s.Q, s.H, s.E, s.q, D, d)
14     else if s = s₂ then
15         y = Judge(s.Q, s.H, s.q, s.d)
16         R.append({"state": s, "output": y})
           //  Transit to the next state.
17         if y starts with "[IRRELEVANT]" then
18             s = s₃(s.Q, s.H, s.E, s.q, s.D)
19         else if y starts with "[RELEVANT]" then
20             s = s₄(s.Q, s.H, s.E, s.q, s.D, s.d)
21     else if s = s₃ then
22         y = NextDoc()
           //  Transit to the next state.
23         if d is NONE then
24             H = s.H + [(s.q, "No Answer")]
25             E = s.E + [s.D[0]]
26             s = s₀(s.Q, H, E)
27         else
28             D, d = s.D + [y], y
29             s = s₂(s.Q, s.H, s.E, s.q, D, d)
30     else if s = s₄ then
31         y = SearchPsg(s.q, s.d)
           //  Transit to the next state.
32         P = y
33         s = s₅(s.Q, s.H, s.E, s.q, s.D, P)
34     else if s = s₅ then
35         y = Answer(Q, H, q, P)
36         R.append({"state": s, "output": y})
           //  Transit to the next state.
37         if o starts with "[UNANSWERABLE]" then
38             s = s₃(s.Q, s.H, s.E, s.q, s.D)
39         else if o starts with "[ANSWERABLE]" then
40             Extract the answer a and the evidence p from y
41             H = s.H + [s.q, a]
42             E = s.E + [e]
43             s = s₀(s.Q, H, E)
44     else if s = s₆ then
45         y = Complete(s.Q, s.E)
46         R.append({"state": s, "output": y})
47         A = y  //  Reach the final state.
48         break
49 return A, R
```

Table 9: Statistics of the warm-up data.

| Module | Branch Token | 2WikiMultiHopQA | Musique | NaturalQuestions | BoolQ |
|---|---|---|---|---|---|
| **Decompose** | **[NEXT]** | 3,500 | 3,500 | 500 | 500 |
| | **[FINISH]** | 500 | 500 | 500 | 500 |
| **Judge** | **[RELEVANT]** | 2,000 | 2,000 | 2,000 | 2,000 |
| | **[IRRELEVANT]** | 2,000 | 2,000 | 2,000 | 2,000 |
| **Answer** | **[ANSWERABLE]** | 500 | 3,000 | 1,500 | 3,000 |
| | **[UNANSWERABLE]** | 500 | 1,000 | 1,000 | 1,000 |
| **Complete** | - | 3,000 | 4,000 | 1,500 | 4,000 |
| *Overall* | - | *12,000* | *16,000* | *9,000* | *13,000* |

## A.4 Construction of warm-up examples

In this section, we elaborate the pipeline to collect training examples for the warm-up stage of AMOR. Given a sample question $Q$ with annotations of the final answer $\hat{A}$, all sub-queries and answers $\hat{H} = [(\hat{q}_j, \hat{a}_j)]_{j=0}^{J-1}$, and all evidence passages $\hat{E} = [\hat{e}_j]_{j=0}^{J-1}$, where $J$ is the number of necessitated sub-queries of $Q$, we construct training examples for four LLM modules of AMOR as follows:

- **Decompose**$(Q, H)$**:** We construct a total of $J+1$ training examples for this module. For each of the $J$ sub-queries, we create an example with the main question $Q$ and the preceding sub-queries and answers $H = \hat{H}_{<j}$ as the input, and the next sub-query $\hat{q}_j$ coupled with the branch token "[NEXT]" as the output (for $j = 0, 1, \ldots, J-1$). Here, $\hat{H}_{<j}$ denotes the sequence containing the first $j$ pairs of sub-queries and their corresponding answers from $\hat{H}$. Additionally, we create one example where the input includes $Q$ and the complete set of sub-queries and answers $H = \hat{H}$, with the branch token "[FINISH]" as the output, indicating the end of the decomposition.
- **Judge**$(Q, H, q, d)$**:** For this module, the input consists of the main question $Q$, the previous sub-queries and answers $H = \hat{H}_{<j}$, the current sub-query $q = \hat{q}_j$, and a document snippet $d$ (for $j = 0, 1, \cdots, J-1$). The output is a branch token that classifies the snippet $d$ as either "[RELEVANT]" or "[IRRELEVANT]" in relation to the current sub-query $\hat{q}_j$. We consider three scenarios for the document snippet $d$: (1) When $d$ is the gold evidence passage $\hat{e}_j$, the output is "[RELEVANT]". (2) When $d$ is a passage from a different document from $\hat{e}_j$, it is marked as "[IRRELEVANT]". We obtain this type of snippet, denoted as $d_j$, by using $\hat{q}_j$ as the query in SearchDoc, ensuring it originates from a distinct document compared to $\hat{e}_j$. (3) When $d$ is a passage from the same document as $\hat{e}_j$ but is not $\hat{e}_j$ itself, it is deemed "[RELEVANT]". We acquire such snippets by invoking SearchPsg with $\hat{q}_j$ to retrieve passages from the same document as $\hat{e}_j$, excluding $\hat{e}_j$ from the results. We refer to this set of passages as $P^-$, considering each of them relevant to $\hat{q}_j$. These varied document snippet scenarios are designed to train the module to discern the relevance of a query to a document based solely on portions of the document content.
- **Answer**$(Q, H, q, P)$**.** Similar to the Judge module, the input for this module comprises the main question $Q$, the previous sub-queries and answers $H = \hat{H}_{<j}$, the current sub-query $q = \hat{q}_j$, and a set of passages $P$ from the same document. The output is either the branch token [UNANSWERABLE]" or a combination of the branch token [ANSWERABLE]", the corresponding answer $\hat{a}_j$, and evidence passage $\hat{e}_j$. We consider two scenarios for $P$: (1) When $P$ does not include $\hat{e}_j$, indicating that the sub-query $\hat{q}_j$ cannot be answered, the output is "[UNANSWERABLE]". Here, $P$ is set to the previously mentioned set $P^-$. (2) When $P$ includes $\hat{e}_j$, suggesting that $\hat{q}_j$ is answerable, we create $P$ by replacing a random passage in $P^-$ with $\hat{e}_j$. For both scenarios, we present the passages to the module in random order when constructing training examples.
- **Complete**$(Q, E)$**.** We construct one training example for this module by setting the input to the main question $Q$ and gold evidence $\hat{E}$ and the output to the final answer $\hat{A}$.

After generating examples from the warm-up datasets using the aforementioned pipeline, we randomly select a specified number of examples. This random sampling aims to ensure a balanced representation of the various modules and branch tokens in the final dataset. Table 9 shows the detailed statistics of the warm-up data.

| Decompose$(Q, H)$ |
|---|

Please continue to decompose the provided main question into answerable sub-queries following previously already solved sub-queries. There are two cases as follows:
(1) [Next] If the question requires further decomposition: Identify and output the next logical sub-query that must be addressed in order to progress towards answering the main question.
(2) [Finish] It means the question does not require further decomposition and can be answered as is.

HERE ARE SEVERAL EXAMPLES:
====Examples Start====
(1) Main Question: What U.S Highway gives access to Zilpo Road, and is also known as Midland Trail?
Output: [Next] How can Zilpo Road be accessed?

(2) Main Question: Which magazine was started first Arthur's Magazine or First for Women?
Solved Sub-Queries:
1. Q: When was Arthur's Magazine started? A: 1844-1846
Output: [Next] When was First for Women magazine started?

(3) Main Question: Which magazine was started first Arthur's Magazine or First for Women?
Solved Sub-Queries:
1. Q: When was Arthur's Magazine started? A: 1844-1846
2. Q: When was First for Women magazine started? A: 1989
Output: [Finish]
====Examples End====

Now Please Complete the Following Task. Please ensure that each sub-query is specific enough to understand in isolation.
Main Question: $\{Q\}\{H'\}$ {%$H'$ is a formatted string representing the solved sub-queries and answers constructed from $H$.%}
Output:

Table 10: Prompt for the Decompose module for HotpotQA.

| Judge$(Q, H, q, d)$ |
|---|

Given a sub-query derived from the main question and a document snippet with its title, please assess whether the document is potentially relevant to the sub-query based on the title and shown content of the document. Assign one of the following two categories:
(1) [Relevant]: Choose this category if the document is relevant to the sub-query.
(2) [Irrelevant]: Choose this category if the document is irrelevant to the sub-query.

HERE ARE SEVERAL EXAMPLES:
====Examples Start====
(1) Main Question: Which magazine was started first Arthur's Magazine or First for Women?
Next Sub-Query: When was Arthur's Magazine started?
Document Snippet: (title: Arthur's Magazine) Arthur's Magazine Arthur's Magazine (1844-1846) was an $\cdots$
Output: [Relevant]

(2) Main Question: Which magazine was started first Arthur's Magazine or First for Women?
Solved Sub-Queries:
1. Q: When was Arthur's Magazine started? A: 1844-1846
Next Sub-Query: When was First for Women magazine started?
Document Snippet: (title: History of women's magazines) In 1693 the first issue of the first women's magazine in Britain $\cdots$
Output: [Irrelevant]

(3) Main Question: What U.S Highway gives access to Zilpo Road, and is also known as Midland Trail?
Next Sub-Query: How can Zilpo Road be accessed?
Document Snippet: (title: Zilpo Road) constructed on the Licking River by the Army Corps of Engineers. $\cdots$
Output: [Relevant]
====Examples End====

Now Please Complete the Following Task:
Main Question: $\{Q\}\{H'\}$ {%$H'$ is a formatted string representing the solved sub-queries and answers constructed from $H$.%}
Next Sub-Query: $\{q\}$
Document Snippet: $d$
Output:

Table 11: Prompt for the Judge module for HotpotQA.

# B Experiments

## B.1 Tool modules

**Tool implementation in AMOR.** We implement both SearchDoc and SearchPsg by adapting Contriever. Given a query, SearchDoc first uses Contriver to retrieve a number of passages from a specific knowledge base and only retains the most relevant passage from each document to serve as the document's representative snippet. Then, SearchDoc returns the top one document snippet and NextDoc can return at most nine more snippets from the remaining ones. On the other hand, SearchPsg returns the top three passages within a given document retrieved using Contriever.

| Answer($Q, H, q, P$) |
| --- |

Please assess whether the sub-query derived from the main question can be answered using the information from the provided passages. Your evaluation should categorize the sufficiency of the information in the passages with respect to the sub-query. Assign one of the following three categories:
(1) [Unanswerable]: Choose this category if the given passages do not contain information to answer it directly.
(2) [Answerable]: Use this category if one of the given passages contains sufficient information to directly answer the sub-query. Provide a clear and concise answer to the sub-query, and the ID of the the corresponding passage.

HERE ARE SEVERAL EXAMPLES:
====Examples Start====
(1) Main Question: Which magazine was started first Arthur's Magazine or First for Women?
Solved Sub-Queries:
1. Q: When was First for Women magazine started? A: 1989
Next Sub-Query: When was Arthur's Magazine started?
Passages: [1] (title: Arthur's Magazine) He was also the author of dozens ···
[2] (title: Arthur's Magazine) Arthur's Magazine Arthur's Magazine (1844-1846) was an ···
[3] (title: Arthur's Magazine) The articles were widely reprinted and helped fuel ···
Output: [Answerable] Answer: 1844-1846; Relevant Passage ID: [2]

(2) Main Question: What U.S Highway gives access to Zilpo Road, and is also known as Midland Trail?
Next Sub-Query: How can Zilpo Road be accessed?
Passages: [1] (title: Zilpo Road) the city which transports people in and out of the city ···
[2] (title: Zilpo Road) Grand Terrace. Access provides public transportation services ···
[3] (title: Zilpo Road) On the other side of the lake is the 700-acre (280 ha) ···
Output: [Unanswerable]
====Examples End====

Now Please Complete the Following Task:
Main Question: {$Q$}{$H'$} {%$H'$ is a formatted string representing the solved sub-queries and answers constructed from $H$.%}
Next Sub-Query: {$q$}
Passages: {P}
Output:

Table 12: Prompt for the Answer module for HotpotQA.

| Complete($Q, E$) |
| --- |

Answer the question ONLY based on the provided passages. Your output should be "yes/no" or a short entity.

HERE ARE SEVERAL EXAMPLES:
====Examples Start====
(1) Question: Which magazine was started first Arthur's Magazine or First for Women?
Passages: [1] (title: Arthur's Magazine) Arthur's Magazine Arthur's Magazine (1844-1846) was an ···
[2] (title: First for Women) First for Women ··· was started in 1989 ···
Output: Arthur's Magazine

(2) Question: What U.S Highway gives access to Zilpo Road, and is also known as Midland Trail?
Passages: [1] (title: Zilpo Road) Zilpo Road ··· can be accessed by Kentucky Route 211 (KY 2112) ···
[2] (title: Morehead, Kentucky) Morehead is a home rule-class city[5] located along US 60 (the historic Midland Trail) ···
Output: US 60
====Examples End====

Question: {$Q$}
Passages: {$E'$} {%$E'$ is a formatted string representing all evidence passages constructed from $E$.%}
Output:

Table 13: Prompt for the Complete module for HotpotQA.

The operation of these tools mirrors the hierarchical interaction paradigm that humans use with search engines [51, 52]: they first identify a relevant document based on short snippets and then refine the search results by focusing within the document.

**Performance comparison across different tool implementations.** On PubMedQA and QASPER, we implement the tools of both AMOR and all baselines using Contriever for a fair comparison. However, for the HotpotQA dataset, the baselines used various approaches for tool implementation, and Table 4 in the main paper directly shows the performance reported in their original papers. To bridge the gap between AMOR and baselines in tool implementations, we conduct experiments to investigate how AMOR performs when using the same tool implementation as two representative baselines, including LUMOS and AgentLM[4], respectively. Please kindly note that LUMOS utilizes GPT-3.5 as a tool for answering questions based on retrieved passages. Therefore, we utilize CPT-3.5 to implement the Complete module when comparing AMOR with LUMOS. Furthermore, the

---
[4]We do not include FIREACT in our comparison due to its reliance on the Google Search API, which exceeds our budget constraints.

Table 14: Comparison results between AMOR and several representative baselines based on L-7B using the same tools on HotpotQA.

| Method | SearchDoc | SearchPsg | Complete | EM |
|--------|-----------|-----------|----------|-----|
| AMOR$_{\text{WFT}}$ | Contriever | Contriever | Fine-tuned L-7B | 30.4 |
| AMOR$_{\text{Process}}$ | Contriever | Contriever | Fine-tuned L-7B | 43.8 |
| LUMOS | Wikipedia API | DPR [18] | GPT-3.5 | 29.4 |
| AMOR$_{\text{WFT}}$ | Wikipedia API | DPR [18] | GPT-3.5 | 31.2 |
| AMOR$_{\text{Process}}$ | Wikipedia API | DPR [18] | GPT-3.5 | **41.4** |
| AgentLM | Wikipedia API | Exact Keyword Match | Fine-tuned L-7B | 22.3 |
| AMOR$_{\text{WFT}}$ | Wikipedia API | Exact Keyword Match | Fine-tuned L-7B | 32.0 |
| AMOR$_{\text{Process}}$ | Wikipedia API | Exact Keyword Match | Fine-tuned L-7B | **43.0** |

Table 15: Datasets for adaptation and evaluation. **Avg. Len** refers to the average length of passages in the corresponding knowledge base, counted by the GPT tokenizer [30]. **Val** is the validation set.

| Dataset | Knowledge Base | Avg. Len | # Train | # Val | # Test |
|---------|----------------|----------|---------|-------|--------|
| **HotpotQA** | Wikipedia Articles | 138 | 2,000 | 100 | 500 |
| **PubMedQA** | PubMed Abstracts | 303 | 401 | 44 | 445 |
| **QASPER** | One NLP Paper | 102 | 700 | 45 | 382 |

adaption to tool implementations of LUMOS and AgentLM necessitates several adjustments in AMOR, including (1) When using "Wikipedia API" to implement SearchDoc, the Decompose module should identify entity names for SearchDoc. (2) When using "Exact Keyword Match" to implement SearchPsg, the Decompose module should predict keywords for SearchPsg. (3) We craft elaborate rules to create warm-up data that follow the above formats for the Decompose module. For example, we use the title of the gold passage as the target entity name for a sub-query and use the longest common string between each sub-query and the corresponding gold passage as the target keyword for "Exact Keyword Match."

As demonstrated in Table 14, AMOR$_{\text{WFT}}$, without being fine-tuned on HotpotQA, surpasses both LUMOS and AgentLM in performance by employing identical tool implementations. Moreover, AMORProcess, upon fine-tuning with process feedback on HotpotQA, exhibits substantial and significant enhancements in performance. These outcomes collectively underscore AMOR's superior performance compared to baseline models and its robustness and adaptability across various tool implementations.

## B.2    Adaptation & evaluation datasets

We describe how we process the datasets as follows: **(1) HotpotQA**: Each document is a Wikipedia article. Since the original test set is hidden, we randomly sample 500 examples from the original validation set for evaluation and split the training set for adaptation fine-tuning and validation. **(2) PubMedQA** [17]: We follow the official split. And we only remain examples whose answers are "yes" or "no" and discard those labeled "maybe." **(3) QASPER** [8]: For each question, we regard the corresponding paper as a knowledge base and each section of the paper as a document with the section name as the title (e.g., "Experiments::Datasets") including several passages. Although many LLMs support context longer than the average paper length of 7k tokens, we focus on testing the ability of language agents to seek and utilize information in this work. We exclude questions that are labeled "unanswerable." Since the original test set is also hidden, we use the original validation set for evaluation and redivide the training set for training and validation. Table 15 shows the statistics of the three datasets.

## B.3    Reasoning process assessment

To investigate the extent to which the adaptation stage enhances AMOR's reasoning process, we conducted a human study with one NLP expert using the HotpotQA test set, Table 16 demonstrates the protocol for annotating the gold feedback $f_{\text{human}}$ and then we calculate the accuracy of the

Table 16: Manual annotation strategy for gold process feedback for different LLM modules.

| Module $m$ | Output $y$ | Gold Process Feedback $f_{\text{human}}$ |
|---|---|---|
| **Decompose**$(Q, H)$ | [NEXT] $q$ | "right", if $q$ is a reasonable sub-query for solving $Q$; "wrong"; otherwise. |
| | [FINISH] | "right", if there are no more sub-queries required; "wrong", otherwise. |
| **Judge**$(Q, H, q, d)$ | [RELEVANT] | "[RELEVANT]", if $d$ is relevant with $q$; "[IRRELEVANT]", otherwise. |
| | [IRRELEVANT] | |
| **Answer**$(Q, H, q, P)$ | [ANSWERABLE] $a\ e$ | "right", if $a$ is the correct answer to $q$ evidenced by $e$; "wrong", otherwise |
| | [UNANSWERABLE] | "right", if $q$ can not be answered based on $P$; "wrong", otherwise |
| **Complete**$(Q, E)$ | $A$ | "right", if $E$ evidence that $Q$ can be answered by $A$; $\hat{A}$, else if $E$ evidence that $Q$ can be answered by $\hat{A}$; "wrong", otherwise. |

Table 17: Accuracy of the silver feedback for four LLM modules based on L-7B.

| Method | Decompose | Judge | Answer | Complete |
|---|---|---|---|---|
| **AMOR**$_{\text{Process}}$ | 81.3 | 95.2 | 84.4 | 82.0 |

Table 18: Proportion of cases where the corresponding error exists. All agents are based on L-7B. N/A means the LUMOS agent does not explicitly execute the relevance judgment step.

| Error Type | LUMOS | AMOR$_{\text{w/o FT}}$ | AMOR$_{\text{Process}}$ |
|---|---|---|---|
| **Format Error** | 5% | **0%** | **0%** |
| **Low Quality Retrieval** | 28% | 18% | **12%** |
| **Question Decomposition Error** | 68% | 60% | **44%** |
| **Relevance Judgment Error** | N/A | **18%** | 20% |
| **Answer Extraction Error** | 46% | 40% | **28%** |
| **Task Completion Error** | 72% | 64% | **48%** |

automatic silver feedback $f$ by comparing it to the gold human feedback. Based on $f_{\text{human}}$, we measured the accuracy of each LLM module's output $y$ (denoted as $\text{ACC}_m$) as follows:

$$
\text{ACC}_m = \begin{cases} 1 & \text{if } f_{\text{human}} = \text{"right"}, \\ 1 & \text{if } f_{\text{human}} \text{ is a refinement of } y \text{ and } f_{\text{human}} = y, \\ 0 & \text{if } f_{\text{human}} = \text{"wrong"}, \\ 0 & \text{if } f_{\text{human}} \text{ is a refinement of } y \text{ and } f_{\text{human}} \neq y. \end{cases} \tag{4}
$$

The accuracy of the reasoning process $\text{ACC}_m$ has been discussed in Table 8 of the main paper. Furthermore, Table 17 presents the accuracy of the silver feedback $\text{ACC}_f$ for AMOR$_{\text{Process}}$. The silver feedback achieves an $\text{ACC}_f$ above 80% for all modules, lending credibility to the use of silver feedback in the adaptation experiments.

### B.4 Error analysis

Table 8 of the main paper has presented a detailed analysis of the accuracy of each module within AMOR on HotpotQA, which suggests the relative weakness of AMOR in question decomposition and task completion. We further conduct a comprehensive error analysis by manually examining the error proportion of different agents. We summarize six error types and show the results of manual annotation in Table 18.

Our thorough analysis accentuates the specific strengths and weaknesses of different agents, unequivocally demonstrating AMOR's relative improvements across key aspects of the complex reasoning process.

Furthermore, we would like to emphasize a significant advantage of AMOR: the ease of diagnosing errors by examining the outputs of its modular architecture. Unlike other agents, where errors are

Table 19: Performance of AMOR parameterized by $\theta_i$ during multi-round adaptation. In the $i$-th iteration ($i = 0, 1, 2$), $\text{AMOR}_{\theta_i}$ is used to explore over the same set of questions or different ones and then is updated to $\text{AMOR}_{\theta_{i+1}}$ based on the exploratory instances.

| Metric | Same Questions | | | Different Questions | | |
|--------|:---:|:---:|:---:|:---:|:---:|:---:|
| | $\theta_1$ | $\theta_2$ | $\theta_3$ | $\theta_1$ | $\theta_2$ | $\theta_3$ |
| **EM** | 45.8 | 45.4 | 45.2 | 45.8 | 45.2 | 45.2 |
| **F1** | 54.9 | 53.4 | 54.7 | 54.9 | 54.5 | 53.6 |

Table 20: Average step/token numbers of different agents. For ReAct and AgentLM, a step refers to a "Thought," "Action," or "Observation" step. For AMOR, a step means a reasoning step within a certain module. And tokens count both input and output tokens.

| Method | Base LLM | HotpotQA | PubMedQA | QASPER |
|--------|----------|----------|----------|--------|
| **ReAct** | **GPT-4** | - / - | 13.4 / 19.0k | 17.5 / 25.3k |
| $\text{AMOR}_{\text{w/o FT}}$ | **GPT-4** | 19.3 / 11k | 9.3 / 7.7k | 10.9 / 6.3k |
| **AgentLM** | **L-7B** | - / - | 11.5 / 7.0k | 12.3 / 8.9k |
| $\text{AMOR}_{\text{Process}}$ | **L-7B** | 20.5 / 4.3k | 11.1 / 2.6k | 11.4 / 2.1k |

often intertwined and obscure (see a ReAct example in Figure 6), AMOR facilitates the provision of precise, module-specific feedback. We will add the above analysis in our revision.

## B.5 Multi-round adaptation

In the main paper, we set $I = 1$ in Algorithm 2 for all experiments, which means that all exploratory instances in the adaptation stage are induced by the warm-up policy $\text{AMOR}_{\text{WFT}}$. We call this setting "single-round adaptation." We are curious about how multi-round adaptation influences the performance of AMOR by adjusting $I$. For the $i$-th iteration ($i = 1, 2, \cdots, I$), we denote the initial parameter of AMOR as $\theta_{i-1}$, which is used to explore over a set of input questions and is updated to $\theta_i$ after exploitation using these exploratory instances. $\text{AMOR}_{\theta_0}$ is exactly $\text{AMOR}_{\text{WFT}}$. During different iterations, we can provide either the same or different questions for AMOR to explore over. The case with the same set of questions is used to simulate an exploration-limited scenario. Note that in this case, the exploratory instances with the same questions are still different due to the ever-changing policy leading to different outputs.

Table 19 shows the performance of AMOR under the multi-round adaptation setting with $I = 3$. We find that the performance is almost unchanged whether using the same or different input questions for each adaptation round. This result suggests that one iteration may be sufficient for the adaptation fine-tuning stage in our study.

## B.6 Token efficiency

Language agents interact with environments to solve problems through frequent calls of LLMs, leading to huge costs in terms of token consumption. Building agents with minimal token usage is essential for curbing deployment costs [46].

Table 20 displays the average number of steps and tokens used by AMOR and ReAct-style agents to answer a question. ReAct-style agents, lacking explicit modeling of inter-step dependencies, require the inclusion of all preceding information in the input for each step. This often results in undesired redundancy. In contrast, AMOR consumes significantly fewer tokens with each module relying only on essential historical information, which highlights the token efficiency of its architecture. When built upon GPT-4, $\text{AMOR}_{\text{w/o FT}}$ uses fewer steps but more tokens than $\text{AMOR}_{\text{Process}}$ based on L-7B due to the additional in-context examples inserted into the prompts of GPT-4.

Table 21: AMOR can be enhanced through targeted fine-tuning and flexibly accommodate additional tools. All results are based on L-7B.

| Method | EM | F1 |
|---|---|---|
| $\text{AMOR}_{\text{Process}}$ | 45.8 | 54.9 |
| +**Targeted Fine-tuning of Complete** | 46.4 | 55.9 |
| +**Additional Tool SearchDemo** | 46.8 | 56.7 |

## B.7 Flexibility

FSM-based reasoning logic is flexible in facilitating targeted enhancements of specific modules and easily accommodating new tools. We conduct two experiments as follows on HotpotQA to demonstrate the flexibility of AMOR, with results shown in Table 21.

**(1) Targeted fine-tuning.** Table 8 reveals that the Complete module of $\text{AMOR}_{\text{Process}}$ still falls short in performance, achieving only $\sim 50\%$ accuracy. We construct 6k examples for the module from the original training set of HotpotQA by treating the final answer $\hat{A}$ as input, and the question $Q$ and evidence passages $\hat{E}$ as output, and then fine-tune the L-7B model on the data. Table 21 shows the performance gains when substituting the original Complete module in $\text{AMOR}_{\text{Process}}$ with this individually fine-tuned L-7B model.

**(2) Accommodating new tools.** Numerous studies have demonstrated the benefits of retrieval-based in-context learning, where a retriever selectively curates tailored demonstrations for each specific input [48]. We implement this by inserting a new state $s_6'$, named "Demonstration Retrieval," before the final state $s_6$ shown in the state transition diagram in Figure 1, making AMOR reach $s_6'$ when Decompose outputs "[FINISH]" at state $s_0$. The new state $s_6'$ holds two variables, including the main question $Q$ and all collected evidence $E$, and employs a tool module SearchDemo to retrieve the top $K$ similar questions to $Q$ from an external demonstration memory, along with their answers and evidence, collectively noted as $\mathcal{K} = [Q_k, \hat{A}_k, \hat{E}_k]_{k=1}^K$. Subsequently, at state $s_6$, the Complete module takes $\mathcal{K}$ as the in-context examples, helping generate the final answer $A$ given $Q$ and $E$. We use the HotpotQA training set as our demonstration memory and employ Contriever-MS MARCO [15] to implement the SearchDemo module, setting $K$ to 5. We fine-tune the L-7B model on the training set to act as the Complete module while ensuring that the demonstration does not include the target question. As Table21 indicates, this integration of such an additional tool further improves $\text{AMOR}_{\text{Process}}$ with targeted fine-tuning.

Additionally, AMOR's reasoning logic can be easily expanded from single-path to multi-path reasoning, akin to the approaches used in Self-Consistency [41], ToT [50], and GoT [3]. This can be achieved by generating multiple outputs within specific modules and incorporating modules that synthesize these multi-path results. Consequently, we advocate for the adoption of the FSM paradigm in the design of future agents. This framework offers the dual benefits of flexibility and the capacity to adapt agents based on process feedback.

## B.8 Case study

We annotate the silver outcome feedback $f_o$ for the Complete module at the final state as $\hat{A}$ if all gold evidence passages are successfully collected (i.e., $\hat{E} \subseteq E$), and "wrong" otherwise.

We demonstrate the advantages of the FSM-based reasoning logic and process feedback mechanism through the comparison between $\text{AMOR}_{\text{Process}}$ and ReAct in Figure 5 and 6, respectively. We observe that ReAct without explicit reasoning logic constraints fails to decompose the question and terminates retrieval prematurely in "Thought/Action 5." Besides, ReAct also mixes right and wrong steps in "Thought 2/4/5," making it challenging for users to critique and improve the agent in a targeted manner. In contrast, AMOR successfully answers the question with a controllable reasoning logic and allows direct process feedback to drive the evolution.

Additionally, Table 22 shows a case where $\text{AMOR}_{\text{WFT}}$ correctly answers a question with the right evidence, yet employs a wrong reasoning process. This underscores the potential unreliability of using

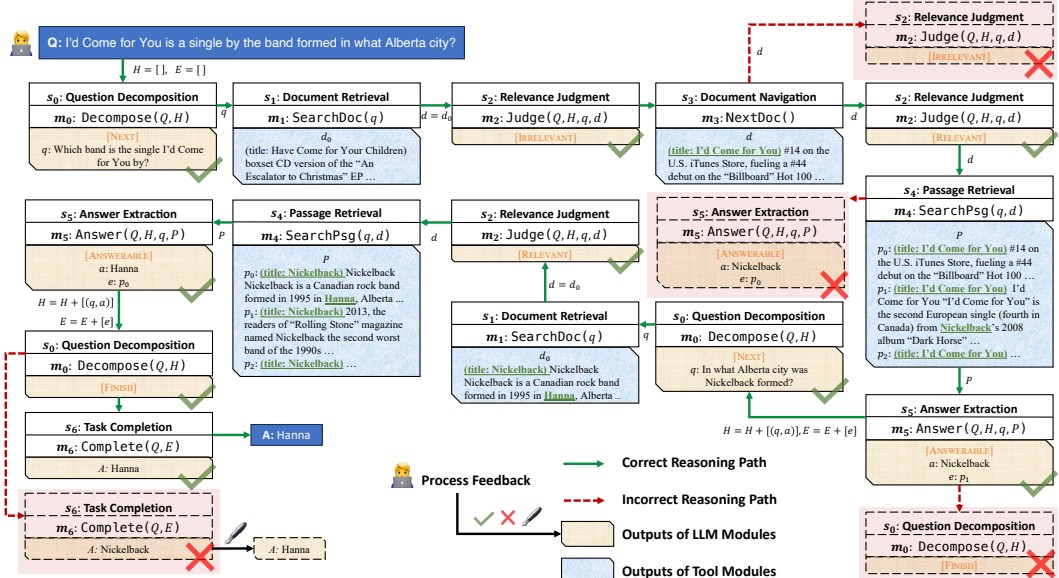

Figure 5: An example demonstrating how AMOR_Process answers a complex question from HotpotQA. Users are allowed to provide direct process feedback to drive the evolution of the agent.

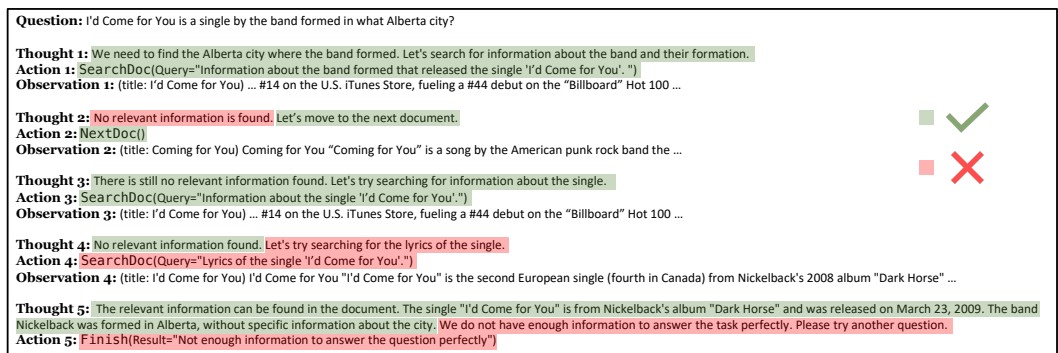

Figure 6: A failure case of ReAct (built upon GPT-3.5) when answering a complex question from HotpotQA by reasoning over retrieved knowledge. The wrong steps are highlighted in red while the right ones are in green.

Table 22: A case where AMOR_WFT answers a question from HotpotQA with the right final answer and evidence but a wrong reasoning process. Each line denotes a reasoning step and shows the output of the corresponding module. The steps highlighted in green are right while those in red are wrong.

**Question: Were the bands Skin Yard and Ostava from the U.S.?**

**Decompose:** [NEXT] Which band has Skin Yard been part in?
**SearchDoc:** (title: Skin Yard) States of America has since reunited, with Finn still drumming · · ·
**Judge:** [RELEVANT]
**SearchPsg:** [1] (title: Skin Yard) States of America has since reunited, with Finn still drumming · · ·
       [2] (title: Skin Yard) Skin Yard Skin Yard was an American grunge band from Seattle, Washington, who were active from 1985 to 1993 · · ·
       [3] (title: Skin Yard) · · · He produced several albums by the grunge bands Soundgarden (a band including original Skin Yard drummer Matt Cameron) · · ·
**Answer:** [ANSWERABLE] Answer: Soundgarden; Relevant Passage ID: [3]
**Decompose:** [NEXT] Which band has Ostava been part in?
**SearchDoc:** (title: Ostava) Ostava Ostava is an alternative rock band from Bulgaria · · ·
**Judge:** [RELEVANT]
**SearchPsg:** [1] (title: Ostava) Ostava Ostava is an alternative rock band from Bulgaria · · ·
       [2] (title: Ostava) of live album. In summer 2006, Ostava supported Ladytron at their second concert in this time at Sunny Beach, Bulgaria · · ·
       [3] (title: Ostava) never officially released a cover version, they do play covers occasionally · · ·
**Answer:** [ANSWERABLE] Answer: Ladytron; Relevant Passage ID: [2]
**Decompose:** [FINISH]
**Complete:** no

outcome feedback to judge the correctness of the reasoning process and the necessity of employing process feedback for adapting agents to specific environments.

## C   Limitations

This study has demonstrated the benefits of two components: (1) explicitly defined FSM-based reasoning logic, and (2) the process feedback mechanism. Nonetheless, a notable limitation must be acknowledged when extending our approach to other tasks. While we have made initial efforts to outline the general principles for crafting the FSM in §3.1 and show the flexibility of adapting AMOR's FSM in Appendix B.7, it still requires a human-driven design process. Looking ahead, our future work aims to enable LLMs to autonomously instantiate FSM-based reasoning logic in Algorithm 1 for diverse user tasks, thereby reducing reliance on human design. Furthermore, we believe that the FSM-based reasoning logic makes it easier for humans to supervise LLMs that potentially outperform humans on the task.

## D   Broader impacts and safeguards

The innovation introduced through AMOR carries significant potential for both positive and negative societal impacts.

On the positive side, the ability of AMOR to adapt to diverse knowledge environments and domains through supervised reasoning could lead to advancements in personalized education, healthcare diagnostics, and customer service. Such applications could democratize access to information and expertise, bridging gaps in knowledge and service availability across different regions and socioeconomic groups.

However, the sophisticated reasoning capability of AMOR also brings about considerations of misuse, such as the generation of disinformation or aiding in the automation of social engineering attacks. Furthermore, if not properly balanced, the tailored knowledge adaptation could unintentionally reinforce biases present in the training data or human feedback, leading to unfair outcomes in decision-making processes that might disproportionately affect marginalized groups.

To mitigate these concerns, it's crucial to engage in transparent development and deployment practices, including bias audits and the establishment of ethical guidelines for use. Additionally, mechanisms for detecting and correcting misinformation or biased reasoning paths should be incorporated into the system's design.

Given the potential for misuse inherent in powerful language models like AMOR, it's vital to implement safeguards. AMOR will be made available under a framework that requires users to agree to ethical usage guidelines before access. Our intention is to maximize AMOR's societal benefits while curtailing the potential for negative impacts, thus ensuring responsible deployment and use of this advanced agent framework.

