# OpenReview forum: "AMOR: A Recipe for Building Adaptable Modular Knowledge Agents Through Process Feedback"
_NeurIPS.cc/2024/Conference — NeurIPS 2024 poster_

### Official Review · Reviewer_bbCe · 2024-07-12

**Soundness:** 3
**Presentation:** 3
**Contribution:** 3
**Rating:** 6
**Confidence:** 4

**Summary:**

This paper proposes AMOR, which is a modular framework for answering questions by reasoning over external knowledge bases. AMOR is designed as a Finite State Machine (FSM), which provides a structured way to break down the question into smaller pieces and solve complex tasks through various steps. Each state in AMOR is a module, which can be an LLM call or a tool call. Each LLM module is fine-tuned separately using data relevant to only that step, thus making data collection and evaluation easier.

**Strengths:**

The authors propose an innovative framework that incorporates FSMs, which is a "classical" technique, with state-of-the-art LLM and tool calling that gracefully incorporates human feedback. The text is well-written and clearly explains all parts of the framework. The figures are also helpful in understanding their proposal. I can see AMOR framework being useful in real-world scenarios like question-answering in private datasets. The paper also presents extensive experiments using popular datasets like HotPotQA and PubMedQA.

**Weaknesses:**

The weaknesses I see are the complexity of this framework: many moving parts, including various LLMs that need to be fine-tuned separately; and the 2 stages fine-tuning approach (warm-up + fine-tuning) makes it even more complex and resource intensive. Dependency on human feedback is another weakness I can see here. However, these are weaknesses common to other agents frameworks so I wouldn't consider this as a disqualifier for this paper.

**Questions:**

I don't have any questions

**Limitations:**

The authors were transparent about possible limitations and negative impacts in the **Broader impacts and safeguards** session and also included suggestions on how to mitigate these problems.

---

> ### Author Rebuttal · Authors · 2024-08-06
>
> > Regarding the complexity and human feedback dependency.
>
> We would like to address your concerns as follows:
>
> - **Regarding the “Separately Fine-tuned LLMs”:** We would like to emphasize that this argument might be inaccurate since we use the same MA-MoE model for all modules and activate different experts to execute their corresponding modules. This approach eliminates the need for separately fine-tuning multiple LLMs.
> - **Regarding the Two-Stage Fine-Tuning:** We argue that the two-stage fine-tuning process is analogous to the widely adopted "pretrain and fine-tune" paradigm. The warm-up stage, which is performed only once, enables AMOR to generalize across different knowledge environments. Once the warm-up stage is complete, AMOR can be deployed and adapted to various domains through the efficient adaptation stage. We believe that the benefits of improved generalization and domain adaptation justify the additional computational cost.
> - **Regarding the Dependency on Human Feedback:** We underscore the importance of maintaining an adaptive mechanism that leverages human feedback to continuously improve performance. Previous agent frameworks often overlook this crucial aspect. AMOR's process feedback mechanism enables efficient and targeted feedback, reducing the overall burden on human supervisors.
>
> In summary, while AMOR introduces complexities, its modular design, two-stage fine-tuning, and process feedback mechanism are deliberate choices that enable it to handle complex tasks effectively and adapt to specific domains efficiently. We believe that the benefits of improved performance and domain adaptability outweigh the concerns raised regarding complexity and human feedback dependency.

---

### Official Review · Reviewer_QRnu · 2024-07-13

**Soundness:** 4
**Presentation:** 3
**Contribution:** 3
**Rating:** 7
**Confidence:** 4

**Summary:**

This work proposes AMOR, a modular approach to building knowledge agents using open-source LLMs. AMOR decomposes tasks into reasoning logic, represented as a finite state machine (FSM) composed of sequentially chained modules. These modules include tools e.g, document retrieval and LLMs, e.g., answer extraction. AMOR retrieves relevant documents for a query and combines information from different sources to produce an answer.

The development process of AMOR includes three stages:
1. **Warm-up Phase:** Each module is fine-tuned on datasets containing not only input-output pairs but also the relevant intermediate steps for each module. This phase ensures that AMOR can generalize across various tasks and knowledge environments.
2. **Adaptation Phase:** The agent is further fine-tuned on specific domains using process feedback. This feedback, which can be human-generated or derived from evaluation benchmarks, is provided at each reasoning step to refine the agent's performance.

Empirical evaluation demonstrates that AMOR effectively utilizes the rich warm-up and evaluation datasets, outperforming relevant baselines. Ablation studies confirm that each component of AMOR is essential for achieving maximal performance.

**Strengths:**

This work is well-motivated and addresses a need for knowledge agents. The modular FSM-based approach is innovative and sensible, allowing for precise reasoning logic. The paper is well-written, and the empirical study is thorough and well-conducted, demonstrating the method's effectiveness through extensive experiments and ablation studies.

**Weaknesses:**

All experiments do not report any measure of uncertainty. Thus, it is impossible to determine if the conclusions are statistically significant. Including measures of uncertainty would strengthen the validity of the results.

**Questions:**

How are the modules chosen in MA-MoE? Is the module/step ID provided to the routers?

**Limitations:**

Yes

---

> ### Author Rebuttal · Authors · 2024-08-06
>
> > Weakness: Regarding the measure of uncertainty.
>
> We agree that it is crucial to provide measures of uncertainty to assess the statistical significance and robustness of the results. To address this concern, we show the mean and standard deviation across three independent runs in the table below.
>
> | Method | Base LLM | HotpotQA EM | HotpotQA F1 | PubMedQA Acc | Qasper EM | Qasper F1 |
> | :--- | :--- | :--- | :--- | :--- | :--- | :--- |
> | AMOR$_{\rm Process}$ | L-7B | $45.8{_{\pm0.25}}$ | $54.8_{\pm0.26}$ | $81.6_{\pm0.33}$ | $19.0_{\pm0.05}$ | $35.4_{\pm0.5}$ |
>
> As shown in the table, the standard deviations across multiple runs are very small, suggesting that our approach yields consistent and robust results. This strengthens the validity of our conclusions and provide confidence in the reported performance.
>
> We will incorporate the uncertainty measures in the revised manuscript to enhance the transparency and credibility of our work.
>
> > Question: Regarding module choosing
>
> Each module corresponds to a distinct expert in the MA-MoE model. When AMOR executes a certain module, its module ID will be provided to the routers of the MA-MoE model to indicate which expert should be activated, thereby enabling our model to be “module-aware.” We will endeavor to provide a more comprehensive explanation of this module-expert mapping mechanism in the revised version of our paper.

---

### Official Review · Reviewer_CkwG · 2024-07-14

**Soundness:** 3
**Presentation:** 2
**Contribution:** 3
**Rating:** 6
**Confidence:** 3

**Summary:**

This work presents a modular pipeline for QA tasks. The pipeline consists of several modules such as question decomposition, document/passage retrieval, answer extraction, etc. Training data is separately constructed for each module (based on existing datasets) and models are individually fine-tuned for the respective modules. This processes is referred to as the 'warm-up' stage. Models are further updated based on feedback obtained during inference using an alignment algorithm (KTO is used in this work). Experiments on QA datasets shows that the proposed method performs better than various baselines in the literature. Several ablations are also provided to study the impact of each component.

**Strengths:**

* The proposed modular approach is sensible and performant.
* Extensive baselines were considered.
* Several ablations were performed that show the impact of various components. It is interesting that the KTO based approach performed better than a simple SFT approach. The human feedback experiment was also interesting.
* Strong performance on three QA benchmarks (HotpotQA, PubmedQA, Qasper).

Overall I think this paper presents interesting ideas. Various presentation aspects can be improved (see weaknesses).

**Weaknesses:**

* The scope for this work was not properly introduced. The paper is motivated from very broad goals only for the reader to later find that the approach and experiments focus on QA tasks. In fact, the paper does not present a formal task description or problem formulation before presenting the methods, leaving the reader to piece together things.
* Method description: In addition to the task not being formally presented, several notations were not clearly defined. For example, equations 1, 3 start talking about a policy which was not described earlier. Although I do understand what's happening in these equations, clearly defined problem and terminology would have made things much clearer.
* In general, I felt the ideas in the paper could've been framed/described in a much simpler way. Readability could be improved.
* The introduction provides motivations but fails to convey any details/intuitions of the proposed method or how it overcomes the challenges mentioned.
* The 'processed based feedback' was not clear to me. I think the paper tries to squeeze in many ideas but this also distracts the reader from the core ideas.
* The experimental setup could have been made clearer, including the various settings such as with/without finetuning and process/outcome feedback.
* Figures and tables are just too small and impossible to parse.

**Questions:**

* What is the without FT setting in Table 6?
* Are the comparisons against baselines fair? Does the proposed approach use more data compared to the baselines?
* What is the major delta of this paper compared to prior modular reasoning works (e.g., Rewoo, Lumos)?

**Limitations:**

I did not find a discussion of limitations in the papers, I suggest the authors to add some discussion.

---

> ### Author Rebuttal · Authors · 2024-08-06
>
> > Weakness 1: Regarding the scope and problem formulation
>
> - **Scope.** AMOR aims to develop a general framework for building adaptable modular LLM agents that can leverage external knowledge sources to tackle complex reasoning tasks. However, we appreciate the reviewer's advice that being more explicit in framing the specific tasks and domains upfront would strengthen our readability. We will clarify the scope early in the next revision.
> - **Problem Formulation.**  We also appreciate the reviewer's advice to provide a clear task formulation upfront. In the revision, we will add a problem statement and formulation section, which will precisely define the QA-style reasoning tasks we use AMOR to solve.
>
> > Weakness 2: Regarding the method description
>
> In Equations 1 and 3, the policy $\pi$ refers to the strategy of the MA-MoE model that maps from the state $s$ to an action $y$.
>
> We will explicitly define it to improve clarity in the revision.
>
> > Weakness 3: Regarding simpler presentation
>
> AMOR uses several novel techniques, including FSM-based reasoning logic, process feedback mechanism, and two-stage fine-tuning strategy. The paper is organized to provide clear definitions, examples, and explanations without oversimplifying the technical details. We will simplify the content in the revision and welcome specific advice from the reviewer on how we could improve the clarity.
>
> > Weakness 4: Regarding the details of AMOR overcoming challenges in the introduction
>
> We believe we have made every effort to convey the key insights and intuitions behind AMOR and how it addresses the stated challenges.
>
> As summarized in Tab. 1, current agents face challenges in three main aspects: **(1):** uncontrollable reasoning logic; **(2):** lack of adaptation mechanisms for new environments; and **(3):** difficulty for humans to intervene in the reasoning process.
>
> In the fourth paragraph (Lines 34-41), we outline the core idea behind AMOR. The FSM-based reasoning logic enables AMOR to solve problems via executions and transitions over a set of modules, allowing for process-based feedback. This design directly addresses **challenges (1)** and **(3)**.
>
> In the fifth paragraph (Lines 42-50), we provide technical details on AMOR's adaptive mechanism, which tackles **challenge (2)**.
>
> We welcome any further feedback from the reviewer on how we could improve the presentation of our work in the revision.
>
>
> > Weakness 5: Regarding process-based feedback
>
> Alg. 1 shows that AMOR solves problems through executions and transitions over a set of modules. In each reasoning step, AMOR executes the module $m$ while in a specific state $s$, obtains the output $y$, and transits to the next state. The overall reasoning process $R$ is formed by a series of such steps. Each step can receive human feedback, termed process-based feedback, as opposed to outcome feedback typically provided for the final step. We will elucidate this concept more explicitly in the revision.
>
> > Weakness 6: Regarding the experimental setup
>
> We provide additional details about the experimental setup below:
>
> - **Without Fine-tuning:** We apply different methods directly to off-the-shelf LLMs without fine-tuning. We provide in-context examples to instruct LLMs to solve given problems. Tab. 12-15 in Appendix A.1 show the prompts for four LLM modules in AMOR under this setting on HotpotQA.
> - **With Fine-tuning:** LLMs are fine-tuned on specific datasets. For AMOR, we employ a two-stage fine-tuning strategy:
>     - **Warm-up Fine-tuning:** The stage fine-tunes an LLM on trajectories from public datasets.
>     - **Adaptation Fine-tuning:** The stage allows AMOR to adapt to specific environments by leveraging different forms of feedback:
>         - **Process Feedback:** AMOR is optimized using feedback provided for both intermediate and final steps of its reasoning process.
>         - **Outcome Feedback:** AMOR is optimized using feedback provided only for the final step of its reasoning process.
>
> We will include these clarifications in our revision.
>
> > Weakness 7: Regarding the table and figure size
>
> We will explore ways to enhance their clarity in the revision.
>
> > Question 1: Regarding the fairness between AMOR and baselines
>
> We have carefully designed our experimental setup to ensure fair comparisons in terms of training data sizes. It involves two groups of comparisons:
>
> - **Comparisons between AMOR$_{\rm WFT}$ and baselines that are NOT fine-tuned on the target datasets:** The table below lists the training data sources and sizes for AMOR$_{\rm WFT}$ and baselines.
>
>
>     | Methods | Data Source | Size |
>     | --- | --- | --- |
>     | FireAct | GPT-4 Trajectories | 2.5k |
>     | AgentLM | GPT-4/3.5 Trajectories | 1.8k |
>     | Self-RAG | Public Resources | 145k |
>     | LUMOS | Public Resources | 57k |
>     | AMOR$_{\rm WFT}$ | Public Resources | 50k |
>     - While FireAct and AgentLM use fewer data than AMOR$_{\rm WFT}$, they rely on the GPT-4/3.5 API for data annotation, which hinders them from scaling up training examples. We believe that AMOR's ability to achieve sota results without any proprietary LLMs is a substantial milestone.
>     - Compared to Self-RAG and LUMOS, which also use public resources, AMOR$_{\rm WFT}$ uses fewer or a comparable number of training examples.
>
>     In summary, this group of comparisons is fair, as AMOR$_{\rm WFT}$ uses fewer or a comparable number of data from public resources without relying on proprietary LLMs.
>
> - **Comparisons between AMOR$_{\rm Process}$ and baselines that are fine-tuned on the target datasets:** AMOR$\_{\rm Process}$, AMOR$_{\rm Outcome}$ and OneR use the same training data from the target datasets, leading to a fair comparison. Table 3 shows the data statistics.
>
> We will provide detailed explanations in the revision.
>
>
> > Question 3: Regarding the difference between AMOR and prior modular agents
>
> Please kindly refer to "Author Rebuttal by Authors" provided at the very beginning.

---

### Official Review · Reviewer_4AJ6 · 2024-07-24

**Soundness:** 3
**Presentation:** 2
**Contribution:** 2
**Rating:** 6
**Confidence:** 4

**Summary:**

This paper proposed an architecture for advanced reasoning in LLMs. The architecture contains several modules dedicated to different tasks in the reasoning flow, each of which can be trained separately using related datasets constructed from public datasets. The proposed method disentangles the reasoning process into sub-steps that are easier to train and suffer less from sparse feedback which is a critical problem for reasoning in traditional LLMs. In the experiments part, the authors conducted sufficient empirical studies, showing the effectiveness of the architecture proposed.

**Strengths:**

1. There's a clear analysis of the bottlenecks that limit the reasoning ability of LLMs. The architecture proposed is designed to solve these problems accordingly. The motivation is clear and the method is reasonable.
2. The structure of the proposed method is well-explained with text and pictures, making it easy for reviewers and readers to understand.
3. In experiments, the choice of baselines has good coverage. The dataset selection respects each subsection's theme, and the experiment setup is reasonable. The experiments are well explained, making it easy to reproduce them.

**Weaknesses:**

1. Currently, many works focus on enhancing the reasoning capability of LLMs. Although the authors mentioned previous works about LLM reasoning and RAG in the related works part, an in-depth explanation of why this method surpasses its predecessors is still needed. A good way to do this is to explain clearly, probably with a few pictures or equations, how those methods dealt with the difficulties and their advantages/drawbacks, and then show that AMOR is better as it avoids some of the existing drawbacks in previous methods.

2. The state transition diagram is well displayed. However, when readers look into the implementation of each module, some key information such as the architecture, hyperparameters, and techniques used for better performance is missing, making it more difficult to understand and reproduce the whole model. A table of these details or a later public code repository would be appreciated.

3. In section 3.3, the detail of how the policy is trained is not well-explained. Did the author use parameterized policies such as policy networks and use a policy-based method in RL for optimization? If so, is the loss presented in equation (3) an auxiliary loss to the loss in policy gradient methods or the pure source of gradients?

4. Following the above question. Will the gradient induced by feedback go through the whole model or only certain blocks?

**Questions:**

See "Weaknesses" part. My concerns listed there are also my questions to the authors.

**Limitations:**

The authors discussed the limitations of their method adequately in the "Limitations" part in the appendix.

---

> ### Author Rebuttal · Authors · 2024-08-06
>
> > Weakness 1: Regarding the difference between AMOR and prior reasoning methods
>
> Please kindly refer to "Author Rebuttal by Authors" provided at the very beginning.
>
> > Weakness 2: Regarding the technical details.
>
> We acknowledge the importance of transparency and reproducibility in scientific research, and therefore we have included the implementation details, including hyper-parameters,  in Lines 204-210 of our manuscript. Additionally, we have submitted the source code for MA-MoE implementation as part of the supplementary material. We intend to release a publicly accessible code repository after the double-blind review period. This repository will facilitate follow-up research to replicate and extend our work.
>
> To further address the reviewer's comments, we plan to include an additional supplementary section in the final version of our paper, which will provide a comprehensive explanation of the algorithm underlying AMOR, elucidating the reasoning process in detail.
>
> We believe that these actions will adequately address the reviewer's concerns, thereby enhancing the clarity and reproducibility of our model.
>
>
> > Weakness 3: Regarding the training details
>
> Equation 3 is the weighted sum of the standard RLHF optimization objective over all modules. Each module corresponds to a parameterized policy network $\pi_{\theta_m}$. The optimization objective is to maximize the reward with a KL divergence penalty. In this paper, we adopt the KTO algorithm [1] for optimization. Equation 3 illustrates the gradients for KTO.
>
> We brief the derivation from Equation 3 to a differentiable loss as follows: As introduced by prior works [1, 2], it is straightforward to show that the closed-form expression for the optimal reward function $r^*_m$ for each module $m$ takes the form: $r^*_m=\beta\text{log}({\pi\_{\theta_m}}/{\pi\_{\theta_m}^{\text{old}}})+\beta\text{log}Z_m$, where $Z_m$ is the partition function. Applying the expression to the Kahneman-Tversky model [1], we can get the differentiable KTO loss. Please kindly refer to the KTO paper for more details. We will provide detailed descriptions to improve the clarity of my work in the revision.
>
> [1] Ethayarajh et al. KTO: Model Alignment as Prospect Theoretic Optimization. ICML 2024.
>
> [2] Rafailov et al. Direct Preference Optimization: Your Language Model is Secretly a Reward Model. NeurIPS 2023.
>
>
> > Weakness 4: Regarding the gradients from the feedback
>
> In the MA-MoE architecture, different modules correspond to different FFN layers and share the same embedding layers and multi-head self-attention layers. Therefore, regarding a training example from $\mathcal{R}_m$ for module $m$, the gradient induced by the feedback will go through the whole MA-MoE model, except those FFN layers corresponding to other modules.

---

> > ### Comment · Reviewer_4AJ6 · 2024-08-08
> >
> > The authors adequately answered my questions and now I have a better understanding of their work. I've raised my grades accordingly.

---

### Author Rebuttal · Authors · 2024-08-06

> Regarding the difference between AMOR and prior reasoning methods

Please kindly refer to the attached pdf file for an illustration of the reasoning processes of AMOR and prior reasoning methods. The table below further elaborates the advantages and drawbacks of prior agents in terms of the following three aspects.

| **Prior Reasoning Methods** | **Reasoning Logic** | **Adaptive Mechanism for New Environments** | **Human Intervention in the Reasoning Process** |
| --- | --- | --- | --- |
| **Retrieval-Augmented Generation** (e.g., Self-RAG) | Sequential Pipeline. **Drawback:** It is difficult to handle complex tasks. | $\times$ (*Undefined*) | $\times$ (*Undefined*) |
| **Agents with Modular Reasoning** (e.g., LUMOS) | *ditto* | Prompting or Imitation Learning from Humans/LLMs. **Drawbacks:** The former often leads to suboptimal results, while the latter suffers from the scarcity of high-quality data. | $\times$ (*Undefined*) |
| **Agents with Free-form Reasoning** (e.g., AgentLM, FireAct) | $\times$ (*Undefined*) | *ditto* | Outcome Feedback. **Drawbacks:** **(1)** Outcome feedback alone is often too sparse and insufficient to improve the intermediate reasoning steps effectively [1]; **(2)** The reasoning steps taken by LLMs can frequently contradict or deviate from the desired outcome [2]. |

In contrast, AMOR addresses the issues of prior reasoning methods in terms of the above three aspects:

**(1) AMOR is equipped with a controllable FSM-based reasoning logic with a stronger capacity for handling complex tasks than simple pipelines employed by Self-RAG, ReWOO, LUMOS.** For instance, if no relevant passages are retrieved from a document, AMOR can dynamically transit to the next document, while LUMOS would be constrained to generate answers based on the irrelevant passages, potentially leading to incorrect or low-quality outputs.

**(2) AMOR adapts to new environments through exploration and exploitation .** AMOR is designed with an adaptation fine-tuning stage, enabling it to adapt effectively to specific domains based on human feedback. This adaptive mechanism sets AMOR apart from prior modular agents that lack the ability to incorporate expert guidance.

**(3) AMOR enables humans to conveniently and effectively intervene and provide feedback at each reasoning step.** AMOR introduces a process feedback mechanism that enables humans to provide direct feedback on the individual modules within the FSM-based reasoning process. This approach facilitates a more natural and interpretable form of supervision, allowing for targeted improvements and fine-tuning of specific reasoning components.

In summary, AMOR achieves more controllable, adaptable, and human-guided reasoning capabilities compared to existing methods. We hope this clarifies how AMOR advances the state-of-the-art in building adaptable and modular knowledge agents. We will include the clarification to better illustrate the proposed method in the final version.

[1] Lightman et al. Let’s verify step by step. 2023.

[2] Liu et al. Score: A framework for self-contradictory reasoning evaluation. 2023.

---

### Decision · Program_Chairs · 2024-09-25

**Decision:**

Accept (poster)

**Comment:**

The authors propose a modular pipeline for QA tasks that performs reasoning over external knowledge bases. The system, AMOR, is designed as a Finite State Machine, whch decomposes complex questions into smaller pieces, then each piece can be handled by different modules such as document retrieval or answer extraction. Each module can be fine-tuned individually, using existing datasets (during warm-up phase); the agent can further be fine-tuned on specific domains using process feedback (via KTO) obtained from either human annotators or an evaluation benchmark.

All reviewers agree that the paper is above the bar. In particular, the reviewers consider this work as well-motivated, addressing important problem; innovative and sensible; experiments are well designed and presented; the method shows strong performance against a good set of baseline systems.

However, multiple reviewers have concerns about the writing, especially the clarity of the submission. To help the authors further improve their work, the reviewers suggested the following:
- To better connect AMOR with prior reasoning methods (like in their rebuttal message to all reviewers).
- Make the KTO objective part clearer.
- Improve the clarity of method description, especially the process-based feedback.
- Make the experiment setup more structured.
- Add error bars.

In my opinion, these suggestions require minor change to the paper, some of them has been done already during the rebuttal discussion. In addition, there is no major concern regarding the method nor the experiments. I believe this work could be found useful to the NeurIPS community so I recommend to accept this paper.